

# Interactions between the atmosphere, cryosphere and ecosystems at northern high latitudes

Michael Boy[1], Erik S. Thomson[2], Juan-C. Acosta Navarro[3], Olafur Arnalds[4], Ekaterina Batchvarova[5,6], Jaana Bäck[7], Frank Berninger[7], Merete Bilde[8], Pavla Dagsson-Waldhauserova[4,9,10], Dimitri Castarède[2], Maryam Dalirian[11], Gerrit de Leeuw[12], Monika Dragosics[13], Ella-Maria Duplissy[1], Jonathan Duplissy[1], Annica M. L. Ekman[14], Keyan Fang[15], Jean-Charles Gallet[16], Marianne Glasius[8], Sven-Erik Gryning[5], Henrik Grythe[11,17], Hans-Christen Hansson[11], Margareta Hansson[18], Elisabeth Isaksson[16], Trond Iversen[19], Ingibjorg Jonsdottir[13], Ville Kasurinen[1,7], Alf Kirkevåg[19], Atte Korhola[20], Radovan Krejci[11], Jon Egill Kristjansson[21,†], Hanna K. Lappalainen[1,12,22], Antti Lauri[1], Matti Leppäranta[1], Heikki Lihavainen[12], Risto Makkonen[1,12], Andreas Massling[23], Outi Meinander[12], E. Douglas Nilsson[11], Haraldur Olafsson[9,24], Jan B.C. Pettersson[2], Nønne L. Prisle[25], Ilona Riipinen[11], Pontus Roldin[26], Meri Ruppel[20], Matthew Salter[11], Maria Sand[27], Øyvind Seland[19], Heikki Seppä[28], Henrik Skov[23], Joana Soares[12,29], Andreas Stohl[17], Johan Ström[11], Jonas Svensson[12], Erik Swietlicki[26], Ksenia Tabakova[1], Throstur Thorsteinsson[13,30], Aki Virkkula[1,12], Gesa A. Weyhenmeyer[31], Yusheng Wu[1], Paul Zieger[11], and Markku Kulmala[1]

[1]Institute for Atmospheric and Earth System Research / Physics, University of Helsinki, P.O. Box 64, 00014 Helsinki, Finland
[2]Department of Chemistry and Molecular Biology, Atmospheric Science, University of Gothenburg, 41296 Gothenburg, Sweden
[3]Earth Science Department Barcelona Supercomputing Center (BSC), Barcelona, Spain
[4]Agricultural University of Iceland, Faculty of Agricultural and Environmental Sciences, Hvanneyri, Iceland
[5]DTU Wind Energy, Technical University of Denmark, Risø Campus, Roskilde, Denmark
[6]Bulgarian Academy of Sciences, National Institute of Meteorology and Hydrology, Sofia, Bulgaria
[7]Institute for Atmospheric and Earth System Research / Forest, University of Helsinki, P.O. Box 27, 00014 Helsinki, Finland
[8]Department of Chemistry, Aarhus University, Langelandsgade 140, DK-8000 Aarhus C, Denmark
[9]University of Iceland, Department of Physical Sciences, Reykjavik, Iceland
[10]Faculty of Environmental Sciences, Czech University of Life Sciences Prague, Czech Republic
[11]Department of Environmental Science and Analytical Chemistry, Stockholm University, S-10691 Stockholm, Sweden
[12]Finnish Meteorological Institute, Climate Research Programme, Helsinki, Finland
[13]University of Iceland, Institute of Earth Sciences, Reykjavik, Iceland
[14]Department of Meteorology and Bolin Centre for Climate Research, Stockholm University, Sweden
[15]Institute of Geography, Key Laboratory of Humid Subtropical Eco-geographical Process, Fujian Normal University, China
[16]Norwegian Polar Institute, FRAM – High North Research Centre on Climate and the Environment, N-9296 Tromsø, Norway
[17]NILU - Norwegian Institute for Air Research, Kjeller, Norway
[18]Department of Physical Geography, Stockholm University, S-10691 Stockholm, Sweden
[19]Norwegian Meteorological Institute, Oslo, Norway
[20]University of Helsinki, Environmental Change Research Unit (ECRU), Ecosystems and Environment Research Programme, Faculty of Biological and Environmental Sciences, P.O. Box 65, 00014 University of Helsinki, Finland
[21]Department of Geosciences, University of Oslo, Oslo, Norway



[22]Tyumen State University, Department of Gryosphere, 625003 Tyumen, Russia
[23]Aarhus University, Arctic Research Center, Climate, Department of Environmental Science, Arctic Research Centre, Frederiksborgvej 399, 4000 Roskilde, Denmark
[24]Icelandic Meteorological Office, Reykjavik, Iceland
[25]University of Oulu, Nano and Molecular Systems Research Unit, P.O. Box 3000, 90014, University of Oulu, Oulu, Finland
[26]Lund University, Department of Physics, Division of Nuclear Physics, P.O. Box 118, 221 00, Sweden
[27]Center for International Climate and Energy Research – Oslo (CICERO), Oslo, Norway
[28]Department of Geosciences and Geography, University of Helsinki, Finland
[29]Air Quality Research Division, Environment and Climate Change Canada, Toronto, M3H 5T4, Canada
[30]University of Iceland, Environment and Natural Resources, Reykjavik, Iceland
[31]Department of Ecology and Genetics/Limnology, Uppsala University, Uppsala, Sweden
[†]deceased August 2016

*Correspondence to*: Michael Boy (michael.boy@helsinki.fi) and Erik S. Thomson (erik.thomson@chem.gu.se)

**Abstract.** The Nordic Centre of Excellence CRAICC (CRyosphere-Atmosphere Interactions in a Changing Arctic Climate), funded by NordForsk in the years 2011-2016, was the largest joint Nordic research and innovation initiative to date, aiming to strengthen research and innovation regarding climate change issues in the Nordic Region. CRAICC

gathered more than 100 scientists from all Nordic countries in a virtual Centre with the objectives to identify and quantify the major processes controlling Arctic warming and related feedback mechanisms, to outline strategies to mitigate Arctic warming and to develop Nordic Earth System modelling with a focus on the short-lived climate forcers (SLCF), including natural and anthropogenic aerosols.

The outcome of CRAICC is reflected in more than 150 peer-reviewed scientific publications, most of which are in the

CRAICC special-issue of the journal Atmospheric Chemistry and Physics. This manuscript presents an overview on the main scientific topics investigated in the Centre and provides the reader a state-of-the-art comprehensive summary of what has been achieved in CRAICC with links to the particular publications for further detail. Facing the vast amount of outcomes we are not claiming to cover all results from CRAICC in this manuscript but concentrate here on the main results which are related to the feedback loops in the climate change-cryosphere interaction scheme affecting the Arctic

amplification.

## 1 Introduction

Near-surface climate warming in the Arctic has proceeded at approximately twice the global average rate since 1980. This extraordinary rate of warming has been recognized since the late 1990s (Serreze et al., 2000) and has accelerated

even since then (Bekryaev et al., 2010), leading to extreme events in 2016 when October-December temperatures in large parts of the Arctic were more than 5˚C above normal and daily anomalies exceeded 16˚C in many locations



(Simpkins, 2017). The warming has caused notable changes in the Arctic cryosphere: Arctic sea-ice area has decreased in all seasons (Wadhams, 2016; Johannessen et al., 2017), glaciers have been retreating (Dowdeswell et al., 1997; AMAP, 2017 and references therein), Arctic freshwaters have rapidly warmed (O'Reilly et al., 2015), Arctic soils and permafrost are warming (AMAP, 2017 and references therein), and precipitation and river discharges into the Arctic

Ocean are increasing (McCelland et al, 2004; Zhang et al., 2013). These changes have dramatic impacts on the ecology and societies of the Arctic and via global climate effects are connected to changes across the planet. Underlining these interconnected changes is the urgent need for a better understanding of the processes contributing to climate change (AMAP, 2017).

It is commonly accepted that increasing concentrations of anthropogenic greenhouse gases predominantly cause the

rising global temperatures and are moderated by effects of aerosols and land-use change (IPCC, 2014). The enhanced warming in the Arctic, termed Arctic amplification or polar amplification, when also considering the Antarctic, is understood to be the result of feedback processes acting specifically at high latitudes and primarily involving sea ice and snow cover (e.g., Bekryaev et al., 2010) and changes in the atmospheric temperature lapse rate, which at high latitudes tends to be opposite to those elsewhere (Pithan and Mauritsen, 2014). The general idea is that an initial

warming (e.g., caused by increased greenhouse gases) reduces the extent of the highly reflective and heat-flux damping snow and ice cover, which results in an increased absorption of solar radiation during summer, and a larger temperature decrease with altitude (lapse-rate) with reduced loss of long-wave radiation, both contributing to further warming (Pithan and Mauritsen, 2014).

The Arctic is more sensitive to the feedbacks described above than the Antarctic, largely because surface air

temperatures in the Arctic are closer to the melting point, at which point the albedo feedbacks are particularly sensitive. However, the relative importance of the different feedback processes (e.g., the role of sea ice vs. snow cover, or the seasonal variation of the amplification) is still debated (Hudson et al., 2011; Perovich and Polashenski, 2012). Furthermore, the implications of Arctic amplification for the mid-latitudes, possibly manifested in more frequent extreme weather events, are under discussion (Cohen et al., 2014).

Other feedback processes specific to the Arctic loom and may be triggered by anthropogenically driven global warming. For example, large reservoirs of carbon are stored in boreal wetlands and permafrost areas. Changes in temperature, water table depth or melting of the permafrost can trigger releases of both methane and carbon dioxide, which can amplify the warming (Chapellaz et al., 1993). Similarly, large amounts of methane are stored in sub-sea permafrost and methane hydrates, which could be destabilized by oceanic warming, although it is not clear how much of that methane

could reach the atmosphere (Shakhova et al., 2010; Myhre et al., 2016). In particular warming summers reduce the volume of multiyear sea ice in the Arctic Ocean that may eventually lead to ice-free summer conditions and a largely changed atmosphere-ocean equilibrium state (Eisenmann and Wettaufer, 2009).

Understanding and quantifying the drivers and complex feedbacks in the arctic and northern boreal climate were the main focus of the CRyosphere-Atmosphere Interactions in a Changing Arctic Climate (CRAICC) Nordic Centre of



Excellence. With support from NordForsk, CRAICC gathered scientists from different disciplines in a virtual Centre of Excellence and used their expertise to obtain a holistic understanding of Arctic feedbacks and interactions. CRAICC involved more than 100 researchers from 24 institutions in the Nordic countries and supported 35 PhD students and postdoctoral researchers. To date, CRAICC has produced more than 150 peer-reviewed scientific publications, and this

paper provides a summary of the main scientific achievements generated by the Centre, with an outlook towards the future. Beyond long-lived greenhouse gases there are other important climate forcers in the Arctic climate system. In particular, short-lived climate forcers (SLCFs) are crucial to Arctic amplification and were one of the main foci in CRAICC.

SLCFs are atmospheric constituents with atmospheric lifetimes (days to weeks) that are substantially shorter than those

of long-lived greenhouse gases (decades). Methane, with an intermediate lifetime of about a decade, is also sometimes considered a SLCF. However, CRAICC concentrated on the substances with the shortest lifetimes, which are primarily different types of aerosol particles, and tropospheric ozone (AMAP, 2015). Importantly, these SLCFs are also air pollutants and in general their climate and air quality impacts must be simultaneously assessed (Stohl et al., 2015; Acosta Navarro and Varma et al., 2016; Acosta Navarro et al., 2017). To understand the impact of aerosol particles on

the Arctic climate, we need to know their sources, including the sources of aerosol precursors such as volatile organic compounds, as well as understand the processes leading to their formation, modification, and removal. We also need to understand how aerosols scatter light and affect clouds, and how albedo and other physical properties of the cryosphere may change due to aerosol deposition. These topics are extremely broad and cover many different scientific scales and disciplines, ranging from biology, to atmospheric science, and to snow physical chemistry and glaciology.

The overall effect of aerosol particles on the global atmosphere is cooling (Myhre et al., 2013), partly because they scatter sunlight back to space. However, some aerosol particles also absorb solar radiation, which warms the atmosphere. Regionally, the net effect depends both on the optical properties of the aerosol species and the reflectivity (albedo) of the underlying Earth surface. For instance, sulphate aerosol particles primarily scatter light, while black carbon (BC) particles as defined in Andreae and Gelencser (2006) and Bond et al. (2013)) are strongly light-absorbing.

Mineral dust is mainly a light-scattering particle in the atmosphere, but always a light-absorbing particle when deposited on snow or ice (e.g., Bond et al., 2013). The importance of light absorption by aerosol particles is enhanced over highly reflective surfaces, such as snow and ice, and therefore the radiative impact of aerosol particles in the Arctic differs from other parts of the planet not covered by snow (Quinn et al., 2008). Furthermore, when light-absorbing aerosol particles, such as BC or dust, are deposited on snow or ice, their warming effect is amplified compared to their

atmospheric impact, because they reduce the snow/ice albedo significantly, which in turn leads to enhanced snow and ice warming and melting (e.g., Flanner et al., 2013). BC has by far the strongest light-absorption and climate warming potential of all aerosol types and is considered globally the second most important climate warming agent after carbon dioxide ($CO_2$) (e.g. Bond et al. 2013). The equilibrium temperature response due to snow darkening by deposited BC is several times greater than that caused by $CO_2$ (Flanner et al. 2007). Thus, BC potentially has a pronounced role in



Arctic warming and melting and as part of CRAICC a particular focus was the investigation of long-term changes in atmospheric BC concentrations and deposition and its climatic impact.

Aerosol particles can also act as condensation nuclei for liquid and ice phase hydrometeors and thus have a profound impact on cloud cover, cloud reflectivity and precipitation (IPCC 2013). Clouds are an important part of the Arctic energy balance, but due to the strong radiative cooling and limited solar insolation, their effects are different than at lower latitudes. Moreover, Arctic clouds are generally very poorly represented in current Earth system models. Further work is therefore required to understand which processes govern the formation, lifetime, and properties of Arctic clouds, including the composition and sources of the cloud condensation nuclei (CCN) and ice nucleating particles (INP) from which these clouds originate. As the Arctic sea-ice melts and more open water is exposed, the emissions of sea spray aerosol, dimethyl sulphide (DMS), and organic aerosols within the Arctic are expected to increase (Nilsson et al., 2001; Struthers et al., 2011; Browse et al. 2014). Such emissions will strongly influence the atmospheric aerosol over the entire Arctic region. Consequently, aerosol components have evident and significant effects on Arctic climate. However, these effects are often complex or even oppositional in terms of their cooling vs. warming effects (e.g. sulphate vs. BC aerosols from same industrial emission sources). The direct and indirect effects of aerosols on climate, and particularly their feedbacks within the cryosphere and high-latitude ecosystems, have hitherto not been well quantified. Such information is essential for a comprehensive assessment of the relative importance of aerosols in high-latitude climate change.

In this paper we discuss the feedbacks affecting the Arctic and boreal zone which were investigated in CRAICC (Chapter 2). In Chapter 3 we introduce the methods applied to analyse the components of the identified feedback loops and interactions. The main results are presented in Chapter 4 where process, interaction, and feedback analysis are discussed. Chapter 5 provides a discussion of the legacy of CRAICC and finally (Chapter 6) we will summarize the outcomes and emphasize necessary future activities.

## 2 Feedbacks affecting the Arctic and boreal zones

Arctic amplification was originally ascribed to the ice-albedo feedback mechanism (Arrhenius, 1896), i.e., initial warming induces melting of highly reflective snow and ice, thus darkening the surface or exposing darker underlying surfaces, with stronger solar absorption properties, which in turn leads to enhanced warming. More recently a suite of causes has been identified as contributing to Arctic amplification (e.g., Serreze and Barry, 2011). These include, but are not limited to, loss of sea ice, changes in atmospheric and oceanic heat flux convergence, changes in cloud cover and water vapor content, and several pathways that link Arctic change to mid-latitude weather (Cohen et al., 2014).

The main goals of the CRAICC project were to quantify the feedback loops identified in the climate change-cryosphere interaction scheme affecting Arctic amplification and pictured in Figure 1. Climate change, cryosphere-atmosphere interactions, and development in society cannot be understood separately but are linked via complex feedback mechanisms. The close cooperation between experts in many scientific areas allowed the CRAICC consortium to





quantify crucial feedback loops, discover a new feedback loop, and to address the potential impact of the components 'Society and human activities' and 'Other feedback mechanisms in the Arctic.' These inputs are compared to the traditional so-called snow/ice-albedo feedback, involving "Forcing", "Arctic Warming" and "Changes in the Cryosphere" (loop A→B→C). In this chapter, we will explain the main feedback mechanisms, which have been further

investigated and partly quantified in CRAICC. Special attention has been paid to studying the processes and feedbacks that are linked to single components in Figure 1 including, e.g., Iceland as a dust-source, the emission of volatile organic compounds from boreal lakes, and the effect of BC deposition on Arctic snow properties.

Within the modern climate, increased burdens of particulate sulphate increase the scattering of incoming solar radiation and thus counterbalance the warming effect of increased levels of greenhouse gases. BC is a species that absorbs

incoming solar radiation and increased BC burdens tend to directly augment the warming effect of greenhouse gases. In addition to direct scattering and/or absorbing effects, airborne particles also have a potentially important effect on atmospheric cloud cover by playing crucial roles in cloud droplet and ice cloud formation. Within CRAICC the feedback loops initiated by airborne particles vis-à-vis changes in anthropogenic emissions (loops D→A→B, D→E→A in Figure 1) have been investigated intensively with cooperation between several research groups.

The climatic impact of dust, and especially that of northern high-latitude mineral dust sources, has received very little attention. Northern high-latitude mineral dust sources, which are often ignored in models and dust effect studies, differ from those in warmer climates and require different parameterization schemes (Bullard et al., 2016). A number of recent studies on high-latitude dust have demonstrated its importance as a substantial contributor to the total Arctic dust load (Bullard et al., 2016; Meinander, 2016; Groot Zwaaftink et al., 2016; Baddock et al., 2017; Wittmann et al., 2017).

However, uncertainties remain large and an assessment of the impacts of high-latitude dust on the cryosphere is still missing. In particular, the impact of climate change on high-latitude dust emissions and potential climate feedbacks are poorly constrained, although reduced snow cover and glacier retreat could both lead to more dust production. The long-term variability of dust events (cf. SYNOP codes from 1949-2011) showed an increase in dust frequency in NE Iceland (towards the Arctic) in 1990s and 2000s (Dagsson-Waldhauserova et al., 2013). Given these open questions, CRAICC

scientists studied the "Dust-Albedo-Feedback-Loop" with a focus on dust emitted from Iceland (loop C→A→B in Figure 1).

Unlike dust, Arctic sea ice has been decreasing in its summer extent and the volume of multiyear ice (Wadhams, 2016; Johannessen et al., 2017). The ice-albedo feedback is partly responsible, but also the warming Arctic land areas and increasing heat input from rivers seem to have added sensible heat into the marine system. As the Arctic sea-ice melt,

exposing more open water and creating more bubbles, the emissions of sea-spray aerosol (sea salt and primary organic aerosols, Norris et al., 2011), dimethyl sulphide (DMS), and secondary organic aerosol precursors within the Arctic are expected to increase (Nilsson et al., 2001; Struthers et al., 2011; Browse et al. 2014), with strong influence on atmospheric aerosol concentrations and therefore likely also on cloud formation in the Arctic region. CRAICC has



investigated the impact of increasing ice-free open water on the emissions of sea-spray aerosol and the influences on aerosol composition and concentration in and for the Arctic region. (loops B→C→A and E→A in Figure 1).

Seasonal snow is another important frozen surface for radiative fluxes in the Arctic, due to its very high albedo. Snow grain size is a primary physical factor defining snow albedo variations (Domine et al., 2006), and air temperature can

affect such snow properties: at the higher temperatures snow grains undergo metamorphosis and become larger (e.g., Flanner and Zender, 2006). Larger snow crystals increase the probability that photons are absorbed, due to the increased optical path within an ice crystal. It results in the enhanced snowmelt and a decrease of the surface albedo which directly leads to stronger absorption of solar radiation. Thus, temperature changes in high latitudes augment positive snow-albedo feedbacks and affect Arctic climate. CRAICC has studied the effect of air temperature on snow albedo using

long-term satellite records on snow cover, surface albedo and air temperature reanalysis data (loop B→C→A in Figure 1).

Aerosol particles and other impurities in snow, including BC, OC, dust and microbes, also affect snow albedo and melt. Snow melt further decreases snow albedo, and an intensive melt can cause the solar zenith angle (SZA) dependent diurnal albedo to become SZA asymmetric (Pirazzini 2004, Meinander et al. 2013). Impurities can also affect snow

physical properties, including density (Meinander et al. 2014), while thick dust layers have been found to insulate, thus preventing snow and ice melt (Dragosics et al. 2016). Therefore, the albedo effect of impurities in snow is best detected at wavelengths where ice absorption is theoretically the smallest and impurity absorption the biggest. For example, some impurities absorb strongly at wavelengths in the UV part of the solar spectrum, (e.g., Peltoniemi et al. 2015), and the absorption by ice is small. The research on albedo decline due to light-absorbing impurities of BC, OC and dust in

snow has been included in CRAICC (loops C&D→A→B in Figure 1).

### 3 Methods to analyse the components of the feedback loops and interactions

CRAICC scientists have used and integrated a large number of state-of-the-art methods covering the spectrum from laboratory studies, to field measurements, and to modelling to quantify the feedback loops presented above. Existing instrumentation and methods have been used and significant new techniques have been developed as part of the

CRAICC project. In this chapter, we give an overview of the most important methods used within CRAICC for elucidating Arctic processes and feedback loops affecting Arctic amplification in the past, present and future.

### 3.1 Historical / paleo data

One useful way to analyse the identified feedback loops and their components is to examine historical records. The climatic history of the Earth includes both distinct warm and cold climatic periods and understanding the climate system

during these past climates can offer insight into present and future feedback processes. One such useful climatic period is the Holocene thermal maximum about 10-5 thousand years BP, when the geologic records indicate that Arctic treeline



advanced to the tundra and Arctic sea-ice extent was reduced compared to the present (Zhang et al. 2017). This period is thus an analogue for the current Arctic greening and sea-ice loss. Such historical records are also useful because they provide a critically important baseline against which current changes can be compared. This is true for climate variables, like temperature, which can be reconstructed from stable isotopes trapped in ice and sediments, and also for many forcing factors (Zhang et al. 2017). In CRAICC, novel studies included compiling long-term BC records, which help to assess whether current BC variations are unique in historical context.

### 3.1.1 Methods to study vegetation-climate interactions in the past

Vegetation plays an important role in mitigating climate change by absorbing atmospheric carbon dioxide, reducing planetary albedo, and influencing the aerosol composition of the atmosphere. Feedbacks between vegetation and climate are particularly conspicuous in the Arctic region, because secondary organic aerosols (SOA) from vegetation constitute a considerable proportion of the atmospheric composition. In addition, increasing vegetation cover, reduces surface albedo and leads to more absorption of solar radiation. This process, called "Arctic greening", has intensified over the past few decades, resulting in increased Arctic plant productivity that is coincident with increases in Arctic surface air temperatures. Tools to study historical vegetation changes include dynamic vegetation modelling, use of plant fossils and other proxy data and proxy-modelling comparisons.

The Lund-Potsdam-Jena General Ecosystem Simulator (LPJ-GUESS) is one important dynamic vegetation model for studying past vegetation-climate feedbacks and an updated version of the LPJ-Dynamic Global Vegetation Model (LPJ-DGVM) (Smith et al., 2001). LPJ-GUESS is one of the most widely used models to simulate past vegetation dynamics from landscape to global scales using a forest gap model scheme (Sitch et al., 2003; Smith et al., 2001). The LPJ-GUESS consists of a number of equations describing the biogeography, biogeochemistry and biophysical processes of ecosystems. The biogeographic features of vegetation are mechanistically represented by plant functional types (PFTs), which are distinguished by different bioclimatic limitations. In the Arctic the boreal forests are generally formed by one or two dominant species, and the PFTs can be individual species, meaning that LPJ-GUESS can simulate vegetation dynamics at the species- or community- levels.

The LPJ-GUESS model not only simulates vegetation growth but also interactions with other components of the climate system, which helps to quantify feedback loops. The model simulates the biophysical and biogeochemical processes of energy and matter exchange between the atmosphere, soil, and biosphere (Hickler et al., 2012; Sitch et al., 2003), which in turn modulate the net primary productivity (NPP), vegetation structure and composition, and the carbon and nitrogen soil and litter budgets, including the soil water. The model has been applied in various investigations, such as carbon cycle studies and investigations of fire occurrence and aerosol changes (e.g., Fang et al., 2015; Schurgers et al., 2009).



Pollen data are the most widely used vegetation proxy data for climate reconstructions. Species apportionment of pollen is used as an indicator of historical vegetation composition, and the chronologies of pollen data are determined by dating surrounding sediments. However, caution must be exercised when analysing pollen data because pollen undergo significant aeolian transport and their presence does not always prove the local presence of the coinciding plant species.

Therefore, it is also useful to use pollen accumulation rate records to investigate species presence and abundance. For example, threshold data above 500 grains/cm²/yr and above 300 grains/cm²/yr are employed as indicators of the local presence of Pinus and Picea forests, respectively (Hicks, 2006; Seppä and Hicks, 2006). Macrofossil and megafossil records are large remains of plants and can be found in small lakes and ponds in the Arctic areas. The presence of such fossil records is a robust indicator of the presence of local sources. Furthermore, due to the carbon composition of

macrofossils and megafossils, their ages can often be determined using radiocarbon dating methods.

Quantitative representations of vegetation structure and function, and interactions with other process-based model components allow for quantifying the feedback loops that include historical vegetation changes. Although proxy-based records are not able to fully quantify the vegetation structure and function, they can provide benchmarks, which can be

used for validating process-based simulations. Furthermore, comparisons between proxy data and model simulations are useful tools for exploiting advantages of both the proxies and the vegetation models. For example, LPJ-GUESS has been utilized to simulate the European Arctic treeline during the Holocene and compared with proxy-based treeline reconstructions. Utilizing an Arctic treeline threshold biomass value of 2 kg C/m² leads to an agreement of the simulated and proxy based treeline, with mismatches seen in mountainous areas (Fang et al., 2013).

**3.1.2 Methods to study long-term records of black carbon (BC)**

Environmental archives, such as ice cores, peats, and lake and marine sediments, chronologically encapsulate material, including material deposited from the atmosphere (e.g., Rose and Ruppel, 2015). These archives can preserve long-term records for up to millennia (e.g., Petit et al. 1999; Zachos et al., 2001); and are essential when assessing past, present and future Arctic climate change, both for setting modern variations into a broader context, and for model

validation. Despite the importance of such data in climate change evaluations, very few records of some important Arctic variables are available (e.g., BC; McConnell et al., 2007). Before CRAICC commenced Arctic BC deposition records were available only from high-altitude Greenland (McConnell, 2010), which does not represent BC deposition in the rest of the Arctic located closer to sea level (e.g., McConnell et al., 2007). BC records are more readily available from Antarctica (e.g. Bisiaux et al., 2011 a and b), the Himalayas (e.g. Xu et al., 2009; Kaspari et al., 2011), and the

European Alps (e.g. Lavanchy et al., 1999; Painter et al., 2013).

Black carbon originates from incomplete natural and anthropogenic combustion of biomass and fossil fuels. Due to myriad emission sources and formation conditions, the term "black carbon" covers a wide spectrum of carbonaceous particles, ranging from charred biomass to soot formed by gas condensation in high-temperature flames. Currently, no



single accepted definition for BC exists, particularly between disciplines (cf., Rose and Ruppel, 2015). Moreover, BC is an operational term which means that its precise definition depends on the method used for its quantification. Coincidingly, no standard BC quantification method exists, and analyses of identical samples have shown measurement methods differ by up to a factor of 7 in concentration (Watson et al., 2005). Schmidt et al. (2001) even reported

concentration differences of a factor of 500 between measured BC in a soil sample inter-comparison study. Thus, comparing results between different methods remains challenging, often particularly between disciplines, as for instance atmospheric measurements may quantify BC particles based on light-absorption properties while snow or soil measurements may extract and quantify BC based on chemical and/or thermal properties (e.g. Hammes et al., 2007). Within CRAICC, two well established and widely used methods were employed for BC analysis from a Svalbard ice

core and four northern Finnish lake sediments. The Svalbard ice core is a 125m ice core collected from the Holtedahlfonna glacier, which covers from year ca. 1700 to 2004. It was analysed using a conventional thermal-optical method for elemental carbon (EC), which is a proxy for BC (Birch and Cary, 1996). After subsampling, ice samples were melted and filtered through quartz fiber filters and EC was quantified with a Sunset Instrument (Sunset Laboratory Inc., Forest Grove, USA) using the EUSAAR_2 temperature protocol (Cavalli et al., 2010) for determining the

carbonaceous aerosol fraction on the filters (Ruppel et al., 2014). Finnish lake sediments were radiometrically ([137]Cs, [210]Pb) dated covering ca. 150 years BP, and were analysed for soot BC (SBC) with a chemothermal oxidation method (CTO-375) developed specifically for BC quantification from sediments (Gustafsson et al., 1997, 2001). After thermal removal of organic material and chemical removal of carbonates from the samples, SBC concentrations were determined with an elemental analyzer (Ruppel et al., 2015). This method detects condensed SBC formed at high

temperatures in gas phase combustion, irrespective of the combusted material (Elmquist et al., 2006). Soot BC particles represent the smallest size fraction of BC, whereas the filter based thermal-optical method used for the ice core samples may most effectively determine bigger char-type BC and agglomerated soot particles.

**3.2 Data from in situ measurements**

**3.2.1 Methods for off-line characterization of particles**

During the last decade, there have been considerable developments with respect to new on-line and off-line techniques, typically based on mass spectrometry, for investigating the chemical composition of atmospheric gases and particles (Nozière et al., 2015; Glasius and Goldstein, 2016, Laj et al. 2009). In this section some of the methods utilized within CRAICC to study atmospheric particles in cryospheric environments and relevant laboratory studies are presented. Further detail of the 'Soot on Snow' project, the sea spray aerosol simulation tanks, and the SMEAR stations is also

given.



Molecular tracers – levoglucosan: Chemical speciation of particles provides information on composition and the processes involved in the formation and growth of those particles. Atmospheric particles are composed of a multitude of organic compounds (Goldstein and Galbally, 2007), and thus it is not feasible to completely elucidate their chemical composition. Instead molecular tracers for specific sources or processes can be identified and investigated. An example

of this is the use of levoglucosan as a tracer for biomass burning emissions in aerosol particles collected on Svalbard in the European high-Arctic (Yttri et al., 2014).

Molecular tracers - secondary organic aerosols: Few studies have explored the formation and distribution of SOA in the Arctic. Hansen et al. (2014) investigated molecular tracers of biogenic and anthropogenic SOA in both North

Greenland and Svalbard using filter collection of particles followed by extraction and analysis by high performance liquid chromatography coupled to quadrupole-time-of-flight mass spectrometry (HPLC/qTOF-MS) using an electrospray ionization inlet. This methodology is well suited for analysis of polar organic compounds often found in oxidized SOA, while less polar constituents, such as the alkanes characteristic of emissions from fossil fuels and their combustion products, are not observed.

Within the last decade organosulphates and nitrooxy organosulphates have been identified as an important, novel class of SOA constituents (Surratt et al., 2007; Iinuma et al., 2007). Organosulphates and nitrooxy organosulphates are analysed using HPLC/qTOF-MS and are identified from the presence of $HSO_4^-$ (m/z = 97) and the neutral loss of $SO_3$ (80 Da) and in the case of nitrooxy organosulphates an additional neutral loss of $HNO_3$ (63 Da, Surratt et al., 2007). The influence of temperature on the gas-particle distribution of semi-volatile compounds is a major challenge to

building a holistic understanding of aerosols in cold climates. Temperatures may change tens of degrees from the ambient air to collection or detection, leading volatile species to evaporate within sampling inlets. This issue requires careful consideration, and potential artefacts should be investigated and avoided using separate gas and particle sampling for off-line analysis (Kristensen et al., 2016).

Inorganic ions and BC in aerosols: Particle size distributions of inorganic anions during Arctic haze were determined using size-selective collection (by MOUDI) followed by ion chromatography (Fenger et al., 2013). Furthermore, long-term monitoring data was used for source apportionment of particles over two years (Nguyen et al., 2013), as well as BC and sulphate (Massling et al., 2015) at the Villum Research Station, Station Nord (81°36' N, 16°40' W), Greenland.

Ice nucleating particles (INP): Cloud processes can also be influenced by ice nucleating particles, those particles which assist in heterogeneous nucleation and growth of atmospheric ice (cf., Vali, 2015). Within CRAICC a combination of particle and surface measurements have been utilized to compare and contrast how quantifiable material and thermo-kinetic properties affect ice nucleation efficiency in relation to the thermodynamic driving force. For studies of the ice nucleation proclivity of particles the CRAICC partners participated in the development of the Frankfurt isothermal



static diffusion chamber for ice nucleation (FRIDGE), and the complimentary electrostatic deposition unit (PEAC7) used to collect particle samples for FRIDGE analysis (Schrod et al. 2016, Thomson et al. 2018). The PEAC7 is a sampling unit for the electrostatic deposition of aerosol particles onto silicon wafer substrates and subsequent characterization in the FRIDGE temperature and humidity-controlled chamber (Schrod et al. 2016). Using high-

resolution photography of the substrate surface ice nucleating particles (INPs) are directly counted as a function of temperature and water vapor saturation. Multiple benefits of the PEAC7 sampling unit include, (i) it enables sampling in clean environments with very low ambient concentrations and (ii) that it enables identification of single INPs and further characterization using scanning electron microscopy. In CRAICC this powerful tool was deployed to two locations to characterize ambient INP concentrations. Those locations included the Villum Research Station and Ny

Ålesund (Svalbard). Each of these locations is characterized by low ambient aerosol particle concentrations and thus suited to the PEAC7 sampling technique.

### 3.2.2 Methods for on-line characterization of particles

Within CRAICC a wide range of instrumentation was used to determine aerosol physical properties in laboratory studies and during field measurements. Some are well established techniques only briefly described below, while others

are more recently developed and thus described in somewhat more detail.

Mass spectrometry: Online quantitative measurements of particle chemical composition for non-refractory sub-micron aerosol particles were performed using an Aerodyne Aerosol Chemical Speciation Monitor (ACSM, Aerodyne Research Inc.; Ng et al. 2011). Gas phase precursors participating in new aerosol particle formation were measured

with several mass spectrometers: including an atmospheric pressure interface time-of-flight mass spectrometer (APi-TOF; Aerodyne Research Inc. and Tofwerk AG) for molecular composition of naturally charged ions and clusters (Junninen et al. 2010); a nitrate chemical ionization atmospheric pressure interface time-of-flight mass spectrometer (CI-APi-TOF; Aerodyne Research Inc. and Tofwerk AG) for neutral clusters like sulfuric acid and organic vapour (Jokinen et al. 2012, Kürten et al. 2014) and a proton transfer reaction time-of-flight mass spectrometer (PTR-TOF;

Ionicon Analytik GmbH) mass spectrometer for organic vapours (Graus et al. 2010). Each technique allows chemical species to be identified by their mass signatures and isotopic fractions (Junninen et al., 2010; Ehn et al., 2010; Schobesberger et al.,2013).

DMPS and SMPS: Aerosol particle size distributions were measured in the field and the laboratory using standard

electrical mobility and optical based techniques. Electrical mobility based instruments rely on size separation of charged particles in a differential mobility analyser (DMA) column. This is followed by condensation of a low volatility liquid on the size selected particles and optical counting in a condensation particle counter (CPC; see for example Wiedensohler et al. 2012 for a detailed description). The combination of a DMA and CPC is referred to as a scanning

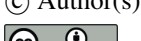


mobility particle sizer (SMPS) or differential mobility particle sizer (DMPS) system. These systems are used for size classification of sub-micron sized particles. For size measurements of larger particles optical particle sizers (OPS) are used.

V-TDMA and H-TDMA: Particle volatility was probed, using thermodenuders in a Tandem DMA setup (V-TDMA), where particles are passed through an oven heated to a known temperature. The heating is followed by a cooling section where gases volatilized from the particles are trapped. Particle size is measured before and after the thermodenuder to assess the contribution of volatile compounds to the aerosol condensed phase. Measurement are performed at a series of oven temperatures to extract information about the volatility distribution of the aerosol constituents. Similarly,

particle hygroscopicity can be ascertained at sub-saturated conditions using a Hygroscopicity Tandem Differential Mobility Analyzer (H-TDMA), wherein particle size is measured before and after exposure to well-defined relative humidity (Liu et al. 1978). Within CRAICC particles in the boreal forest environment were also characterized using a combination of the two instruments (i.e. the VH-TDMA; Hong et al. 2014).

CCNC: The ability of particles to form cloud droplets was measured using a Cloud Condensation Nucleus Counter (CCNC) of the continuous-flow thermal-gradient diffusion type (Droplet Measurement Technologies). The CCNC operates by exposing particles to well-known supersaturations of water, generated by applying a temperature gradient over a wetted column, and using optical detection to monitor droplet formation (Roberts and Nenes, 2005).

PINCii: Instrumentation for online analysis of INP was developed as a result of a formalized technology sharing agreement and collaboration by six institutional partners (Ulrike Lohmann group, ETH-Zurich; Frank Stratmann group, TROPOS-Leipzig; Markku Kulmala group, University of Helsinki; Merete Bilde group, Aarhus University; Erik Swietlicki group, Lund University; and the University of Gothenburg), that was initiated by the CRAICC partners. The six-partner group has worked to develop and build a next generation Portable Ice Nucleation Chamber (PINCii, Figure

2), which is a continuous flow diffusion chamber (CFDC) designed to update earlier parallel plate CFDCs (e.g. ~ZINC, PINC, SPIN; cf., Stetzer et al. 2008, Garimella et al. 2016). The PINCii instrument is an ice-coated flow tube reactor system designed to stimulate and measure ice nucleation within a test aerosol flow. Dry particles sampled from ambient or laboratory generated flows are injected into a chamber, which contains a controlled water vapour supersaturated environment with respect to ice. Thus, by monitoring both the input and output flow the fraction of INP can be directly

determined.

### 3.2.3 SoS-project and Sea spray aerosol simulation tanks

Soot on Snow (SoS) project: As part of CRAICC, the SoS project was conducted to study the effect of light-absorbing particles on snow surfaces. It consisted of a series of field experiments where BC and other light-absorbing impurities,



including Icelandic dust, were dry deposited onto the surface of natural snow packs and the consequent effects on albedo, snow density, other physical properties, including melting were measured during the spring season. The broadband albedo was measured with pyranometers, in addition to the directional reflectance of snow. Concentrations of EC in the snowpack were analyzed using a thermal-optical method (as described for the Svalbard ice core, Sec.

3.1.2) and compared with the measured albedo and that modelled with the SNow, ICe, and Aerosol Radiative (SNICAR) model (Flanner et al., 2007; 2009). Details of the experiments were presented by Meinander et al. (2014), Peltoniemi et al. (2015), and Svensson et al. (2016).

Sea spray aerosol simulation tanks: During CRAICC significant effort has been devoted to improving the understanding

of sea spray aerosol. One important development has been the design, construction, and use of new temperature-controlled sea spray aerosol simulation tanks (King et al. 2012, Salter et al. 2014). In these tanks air is entrained in real or artificial seawater via frits/diffusers or via plunging jets. The entrained air breaks up into bubbles which rise to the surface where aerosols are generated by bubble bursting processes. These tanks can be coupled with other aerosol characterization instrumentation, for example particle size and number concentrations (see above), and thus can be used

to probe a variety of physical and chemical properties of sea spray aerosol, including cloud forming ability, hygroscopicity, and volatility.

### 3.2.4 Long-term measurement stations involved in CRAICC

The CRAICC core permanent research infrastructures included 18 well-established field research stations, covering ecosystems from Arctic to boreal locations (Figure 3). These stations provided the time-resolved data sets used by

CRAICC community, and are reflected in many publications from the Nordic Centre of Excellence.

### 3.3 Multiscale modelling

Different modelling systems have been utilized by CRAICC to simulate myriad levels of Earth systems. The main tools used in the consortium are described in detail below.

### 3.3.1 Process-based modelling of the formation and growth of SOA in the Arctic region

The CRAICC consortium contributed to the continuing development of a 2-dimensional Lagrangian Aerosol Dynamics, gas and particle phase CHEMistry and radiative transfer model (ADCHEM; Roldin et al., 2011) with improved representations of the biogenic secondary organic aerosol formation (Hermansson et al., 2014, Öström et al., 2017). New process-based schemes for aerosol dynamics, particle-phase molecular diffusion mass transfer limitations, organic- and inorganic particle-phase chemistry and gas-phase chemistry schemes were implemented and constrained

based on laboratory smog chamber experiments (Roldin et al., 2014, 2015). The latest version of ADCHEM includes





a detailed gas-phase chemistry scheme that is based on the Master Chemical Mechanism (MCMv3.3.1, Jenkin et al., 1997; Saunders et al., 2003). This scheme also includes a novel scheme for the formation of highly oxygenated organic molecules (HOMs) formed from ozonolysis and OH-oxidation of monoterpenes. The HOMs formation scheme is based on experimental work by Ehn et al., (2014) that was recently used to evaluate the contribution of HOMs to the activation

and growth of new particles during observed new particle formation events in sub-arctic forests (Öström et al., 2017). The non-equilibrium SOA formation scheme simulates the size resolved particle growth using concentrations of around 700 different organic molecules, provided by the gas-phase chemistry scheme. The SOA scheme can also account for heterogeneous oligomerization and non-ideal organic and inorganic particle-phase interactions as well as the impact of particle-phase mass transfer limitations on the formation and evaporation of SOA particles (Roldin et al., 2014, Roldin

et al., 2015, Öström et al., 2017).

### 3.3.2 Meso-scale modelling of Arctic BC and Icelandic dust deposition

To assess long-term BC concentrations and deposition in the Arctic an offline Eulerian chemical transport model was run for the period between 1980 and 2015. The System for Integrated modeLling of Atmospheric composition (SILAM), is documented in detail by Sofiev et al. (2006, 2014). The SILAM model has several chemical transformation

modules, including gas-phase chemistry and secondary inorganic aerosol formation, linearized sulphur oxide chemistry, radioactive nuclide decay, and aerosol dynamics (condensation and coagulation) computed either from thermodynamic equilibrium or as dynamically. The aerosol size spectrum is described with a sectional approach with a user-defined bin distribution. Mechanisms of dry deposition vary from primarily turbulent diffusion driven removal of fine aerosols to primarily gravitational settling of coarse particles, depending on the particle size (Kouznetsov and

Sofiev, 2012). Wet deposition distinguishes between sub- and in-cloud scavenging by both rain and snow (Horn et al., 1987; Smith and Clark, 1989; Jylhä, 1991; Sofiev et al., 2006). The dispersion model considers BC as an inert pollutant, with a size distribution described by a single size bin ranging from 0.001 to 1 $\mu$m in dry particle diameter. Total particle production is integrated over each size bin (non BC aerosol utilizes full bin distribution) while dry and wet removal rates are calculated using mass-weighted mean diameter in each size bin. The SILAM model has been extensively

evaluated against European and global air quality observations (Solazzo et al. 2012, Huijnen et al. 2010, Ruppel et al., 2017), and is driven by ERA-Interim (Dee et al. 2011) meteorological data with 3-hour temporal and 0.72 ° horizontal resolutions. SILAM uses the MACCity emission dataset (Granier et al., 2011) for anthropogenic emissions, except for flaring emissions, which were taken from the ECLIPSE dataset (Stohl et al, 2013). Emissions are available every five years, begining in 1980 for MACCity and in 1990 for ECLIPSE, with the remaining years estimated by linear

interpolation. Global simulations utilized a horizontal resolution of 0.72° × 0.72° and a vertical grid consisting of 9 unevenly spaced atmospheric layers. The lowest, thinnest layer was 25 m thick, with the top layer reaching into the





stratosphere. The source contribution to BC deposited at Holtedahlfonna was investigated by tagging the different emission sectors while computing atmospheric dispersion of BC.

To support the interpretation of dust storm observations in Iceland, simulations with the numerical weather prediction model Hirlam (Unden et al., 2003) were used. The simulations were run on a 5 km horizontal grid at 65 vertical levels,
with forcing at the boundaries from the operational suite of the ECMWF. The turbulence calculations were based on the CBR scheme (Cuxart et al., 2000) and the Interaction Soil Biosphere Atmosphere (ISBA) employed at the surface (Noilhan and Mahfouf, 1996).

### 3.3.3 Earth System Models (ESM) applied for feedback simulations

NorESM1-M is the Norwegian Earth System Model version 1 (Bentsen et al., 2013; Iversen et al., 2013). This particular
set-up of NorESM1, used in both CRAICC and in the Coupled Model Intercomparison Project Phase 5 (CMIP5), has an intermediate horizontal atmospheric resolution of $1.9° \times 2.5°$ and 26 vertical levels. The ocean module is an updated version of the isopycnic ocean model MICOM, while the sea ice (CICE4) and land (CLM4) models and the coupler (CPL7) are similar to CCSM4 (Gent et al., 2011), differing only in the tuning of snow grain size for fresh snow on sea ice within CICE4. The atmosphere module, CAM4-Oslo (Kirkevåg et al., 2013), is a version of CAM4 extended with
advanced representation of aerosols, aerosol-radiation and aerosol-cloud interactions.

The land model has its own carbon cycle model, and includes the SNICAR model (Flanner et al, 2007; 2009). The latter facilitates calculations of the effects on radiative transfer from snow darkening by deposited light-absorbing aerosols, i.e. black carbon (BC) and mineral dust. The albedo effect of light-absorbing aerosols deposited on snow-covered and bare sea ice is also accounted for in the sea-ice model.
CAM4-Oslo calculates mass concentrations of aerosol species that are tagged according to production mechanisms in clear and cloudy air for four size-classes (nucleation, Aitken, accumulation, and coarse mode). These processes are primary emission, gaseous and aqueous chemistry (cloud processing), gas to particle production (nucleation), condensation, and coagulation of small particles onto larger pre-existing particles. Loss terms are dry deposition, in-cloud and below-cloud scavenging. Aerosol components included are sulphate, BC, organic matter, sea-salt, and
mineral dust, distributed over 11 modes for externally and internally mixed particles that are emitted or produced in air and 9 which are tagged to production mechanisms in air or in cloud droplets. In addition to these 20 transported tracers for aerosols, there are two additional tracers for the gaseous aerosol precursors $SO_2$ and DMS. Optical properties are estimated from Mie-theory, while super-saturation and hygroscopic growth for CCN-activation are calculated from Köhler theory.
Internally mixed water from water vapour condensation is treated separately using look-up tables for the aerosol optical parameters. Other look-up tables are used to obtain dry size parameters (dry radii and standard deviations) of the aerosol population, which are used as input in the calculation of CCN activation, following Abdul-Razzak and Ghan (2000).





A separate post-CMIP5 NorESM1 version with explicit parameterizations of nucleation and secondary organic aerosols (Makkonen et al., 2014) has also been used in CRAICC. In this model version, hereafter referred to as NorESM1-CRAICC, the land model uses the MEGAN parameterization for interactive biogenic volatile organic compound (BVOC) emissions, instead of the prescribed sources applied in the CMIP5 version of the model.

The atmosphere model CAM4-Oslo has also been extensively validated and compared with other models through the AeroCom initiative (Aerosol Comparisons between Observations and Models: aerocom.met.no), by Jiao et al. (2014), Tsigaridis et al. (2014), Kipling et al. (2016), and Koffi et al. (2016). A separate evaluation of NorESM1-M and other CMIP6 models by use of remote sensing of aerosols in the Arctic has been made by Glantz et al. (2013). NorESM1-M is also taking part in the ongoing Precipitation Driver Response Model Intercomparison Project (PDRMIP: see e.g.

Samset et al., 2016; Myhre et al., 2017).

**3.4 Data collected through remote sensing techniques**

**3.4.1 Aerosol lidar (CL51 ceilometer)**

The atmospheric boundary layer in the Arctic is of key importance because it connects the atmosphere, cryosphere, and marine components of the climate system. Nevertheless, observational data from the Arctic boundary layer is sparse.

This is partly due to the hostile climatic conditions with low and even extreme temperatures that are beyond the operational range of many meteorological instruments. Long-term climatological measurements are traditionally performed with rugged instruments such as sturdy cup-anemometers and wind-vanes located near the ground – typically at heights between 2 and 10 m. Except for the few sites with regular radio-soundings, limited information on the vertical structure of the Arctic atmospheric boundary layer is available.

A preponderance of the information on the structure of the Arctic atmospheric boundary layer has been obtained during intensive observations carried out over limited time periods, primarily during summer months (Gryning et al. 1985; Lambert et al. 2011; Burgemeister et al. 2013, Di Liberto et al., 2012; Batchvarova et al. 2014; Achert et al., 2015). However, recent technological improvements of ground based lidar remote sensing may significantly benefit future Arctic meteorological research. Utilizing components developed for use in fiber optics communication, lidars have

become more compact, reliable, and easier to use. The development of eye-safe fiber-based lidars started in the mid 1990's and the first commercial products became available around 2005. One such instrument, an aerosol lidar (CL51 ceilometer from Vaisala) was installed at Villum Research Station in spring 2011. The CL51 measures the backscatter profile of the aerosols in the atmosphere, and has survived to measure in the harsh Arctic conditions for several years. The aerosol backscatter is measured as a function of height with a vertical resolution of 10 m up to 7 km height. The

backscattered signal depends on the number concentration, size, and optical properties of the particles in the air.





### 3.4.2 Satellite

Satellites provide a plethora of environmental data, including information on atmospheric composition and surface properties over wide spatial areas. Their instruments provide coverage that is spatially and temporally delineated based on orbiting patterns and swath width etc. Satellite observations are highly complementary to ground-based *in situ* and

remote sensing measurements, and they make observations possible over remote and difficult-to-access areas where *in situ* measurements are not available. However, satellites are typically confined to optical (UV/VIS), infrared (NIR/TIR) and microwave (Radar) observations that do not necessarily contain the same detail as ground-based measurements. For atmospheric composition, different instrument platforms are used to detect trace gases, greenhouse gases, and aerosols and clouds. For many optical techniques satellite data are only collected when solar radiation is available and

solar zenith angles are large enough, which is a serious limitation over polar regions.

The detection of gases requires high spectral resolution (spectrometers, interferometers) with specific wavelengths for the gases of interest. The design of such instruments implies a relatively low spatial resolution of tens of kilometres. Examples are SCHIAMACHY, GOME-2, OMI and TROPOMI, with in incremental improvement of the spatial resolution which for the recently launched (2017) TROPOMI instrument on the sentinel-5P satellite is 7x7 km2 sub-

nadir as compared to the OMI footprint of 13x24 km2. TROPOMI extends the capabilities of OMI and measures column concentrations of ozone, formaldehyde, carbon monoxide, $NO_2$, $SO_2$ and methane, as well as aerosol layer height using the O-A band and the absorbing aerosol index, a qualitative parameter indicating the presence of absorbing aerosol particles, and UV radiation. Information on the trace gases such as $NO_2$, $SO_2$ and formaldehyde (a proxy for less volatile organic compounds) and near-surface UV radiation is also important for the formation of aerosol particles

through gas-to-particle conversion.

Dedicated instruments for quantitative measurements of aerosol and cloud properties are radiometers with moderate spatial resolution which do not require high spectral resolution but need a wide range of spectral bands (e.g. MODIS, VIIRS), and preferably also several viewing angles (MISR) and polarization (POLDER). Other instruments which are used for the retrieval of aerosol properties are, e.g., SeaWiFS, MERIS, the AVHRR instruments which together provide

a long-time series which started around 1981 and the dual view ATSR series (ATSR-2, AATSR; 1995-2012) followed up by SLSTR launched in 2016 on Sentinel-3.

Some of the instruments mentioned above are also used, or even designed, to retrieve information on land and ocean surface properties. The AVHRR, ATSR and SLSTR instruments were designed to measure land and ocean surface temperature. MODIS also provides land surface temperature, land surface albedo and BDRF and several of the sensors

mentioned provide information on forest fires by virtue of their TIR channels. SeaWiFS, MERIS and OLCI (on Sentinel-3) were designed to measure ocean parameters and in particular ocean colour.

Satellite information has been used in the CRAICC feedback analysis, as described in more detail in Sect. 4. The use of satellites to study spatial changes of glaciers, sea-ice extent, dust storm occurrence and deposition is discussed in



Sect. 4.1.1.1. High spatial resolution sensors such as Landsat and Sentinel-2 provide detailed information on land surface properties and were used to detect high latitude dust sources and the spatial extent of glaciers. MODIS was used to detect dust storms over Iceland.

Two Moderate Resolution Imaging Spectroradiometers (MODIS) onboard the Terra and Aqua satellites have been
routinely collecting information on multiple environmental parameters since the year 2000. CRAICC studies on large-scale snow-covered surface albedo changes were based on MODIS products on snow cover fraction and surface albedo (MCD34C3). The MODIS snow mapping algorithm is based on the Normalized Difference Snow Index (NDSI), which utilizes the fact that snow has a high reflectance in the visible part of the solar spectrum and low reflectance in the infrared (Hall et al., 2002). The MODIS Bidirectional Reflectance Distribution Function (BRDF) and Albedo algorithm
utilizes multiple spectral bands to retrieve broadband albedo information (for detail see Lucht et al., 2000; Schaaf et al., 2002). MODIS data products are consistently validated, readily available and widely used by research community, including CRAICC.

MODIS data have been used by Atlaskina et al. (2015) to study the temperature dependence of the albedo of snow covered land surfaces in the northern hemisphere as described in Sect. 4.2.3. MODIS and AATSR have been used to
explore the retrieval of aerosol parameters in Arctic regions (Istomina et al., 2010; Mei et al., 2013a, 2013b). For an overview of aerosol remote sensing in polar regions, using ground-based and satellite instruments, see Tomasi et al. (2015). The use of satellite data to study Arctic Amplification has also led to a controversy in the literature as described in Sect. 4.2.2.

## 4. Process, interaction, and feedback analysis

The CRAICC research package focused on the identification and quantification of high-latitude Earth system feedbacks. This required a holistic understanding of essential Arctic systems and their interactions, and included long-term observations and detailed measurement campaigns as well as complementary multiscale-modeling platforms. This multi-pronged research strategy allowed the Centre to assess myriad pathways outlined within the diagram of Arctic feedbacks (Figure 1). Individual studies have focused on particular components (A, B, C, D, E in Figure 1), or
interactions between two or more components, including attempts to integrate and quantify fully connected feedback loops. That said, given the large natural variability of Arctic systems and significant coupling interactions with lower latitudes, the quantification of single feedback loops and parameters remains challenging.

In general, quantifying a feedback loop requires the observation of the dampening (negative) or strengthening (positive) of a system perturbation. For the Arctic human activities are one trigger for environmental change, although CRAICC
has also considered Arctic feedbacks in the absence of anthropogenic forcing. In this chapter, we present the main outcomes of CRAICC research related to changes in natural emissions and processes (sub-chapter 4.1 to 4.4) and changes in the Arctic based on anthropogenic impacts (sub-chapter 4.5). However, the reader should remember that in total CRAICC published more than 150 manuscripts in international journals and here only a number of selected results




are highlighted with reference to original publications for more detail. A list of all CRAICC publications is available at the project website: https://www.atm.helsinki.fi/craicc.

### 4.1 Atmosphere

### 4.1.1 Icelandic deserts and dust (component E in Fig. 1)

Iceland straddles the Arctic circle in the north Atlantic Ocean and thus is significant as a source and monitoring point for Arctic climate change. That said *in situ* aerosol observations from Iceland are minimal. The City of Reykjavik and the Environmental Agency of Iceland have maintained Particulate Matter ($PM_{10}$; diameter less than 10 $\mu$m) observations at a couple of fixed locations for <15 years in Reykjavik, far from the local dust sources (Thorsteinsson et al., 2011), while shorter term monitoring installations have been established during volcanic eruption events

(Thorsteinsson et al., 2012; Dagsson-Waldhauserova et al., 2014a). There also exists a unique dataset of dust and volcanic ash observations that have been collected by the Icelandic Meteorological Office for nearly a century. Frequent volcanic eruptions increase the annual dust frequency and the consequent dust-volcanic ash resuspension events prolong the impacts of eruptions (Thorsteinsson et al., 2012; Thordarson and Höskuldsson, 2015). For example, after the Eyjafjallajokull eruption in 2010, the erosion flux exceeded 11 000 kg m$^{-1}$ during one dust event (Arnalds et al.,

2013), which is one of the most severe wind erosion events ever recorded on Earth.

Given its location and ecosystem much of the aerosol particulate in Iceland originates from wind-surface interactions. Icelandic surfaces are classified by the ''AUI Nytjaland" land cover database into vegetation classes (Gisladottir et al., 2014), which shows that Iceland's total desert area is about 43,400 km$^2$ (Arnalds et al., 2016). About 15,000 km$^2$ of the total desert are active aeolian surfaces, which include super active dust hot spots. Over the long-term Iceland has been

observed to have 34-135 dust days per year without strong seasonality (Dagsson-Waldhauserova et al., 2013, 2014a). Dust is produced at a rate of about 31-40 million tons per annum and that dust is redeposited onto the land, into the ocean, and onto glaciers (total area of > 500,000 km$^2$), with some dust plumes capable of traveling over 1000 km (Figure 4, MODIS data; Arnalds et al., 2016). Iceland is the largest Arctic and European desert with a dust event frequency comparable to major lower latitude desert areas like the Gobi or Iranian deserts (Arnalds et al., 2016).

The lack of established observations led CRAICC to support several field-dust-aerosol measurement campaigns between 2013 and 2016 (Dagsson-Waldhauserova et al., 2014b, 2015, 2016). During severe dust storms $PM_{10}$ concentrations exceeding 6,500 $\mu$g m$^{-3}$ were measured (1 min. averaging time) with median $PM_{10}$ values > 1,000 $\mu$g m$^{-3}$ during 24-hour intervals. During a moderate storm a $PM_1$ maximum of 261$\mu$g m$^{-3}$ was measured. High values of submicron particles are typical of Icelandic volcanic dust, with resulting $PM_1/PM_{2.5}$ ratios of > 0.9, while $PM_1/PM_{10}$

ratios range from 0.34 to 0.63. These values are comparable to urban air pollution rather than natural dust storm events observed in other regions. Particle number concentrations (PM~0.3-10 μm, Optical Particle Sizer) are also high, with maximum concentrations of ~150 000 particles cm$^{-3}$ recorded with a primary peak in the size distribution between 300



and 337 nm, and a smaller peak for particles with diameters between 1.5 and 5 µm. Such extreme concentrations are reflective of ongoing volcanic eruptions. The dust in Iceland is primarily volcanic in origin and dark in colour, with many particles that include sharp-tipped shards and large bubbles. About 80% of PM is volcanic glass that is rich in heavy metals, such as iron and titanium.

Satellite data collected since the 1970s yields detailed information on spatial changes of glaciers, sea-ice extent, dust storm occurrence, and dust deposition in Iceland. High resolution imagery from the LANDSAT-8 (NASA) and SENTINEL-2 (ESA) satellites allows for improved analysis of dust source areas, although at latitudes above 65° temporal resolution is limited to approximately twice weekly. Despite having a relatively mild and humid climate dust storms are visible in Iceland about 40 days a year on average (MODIS 250 m resolution images), which is a conservative

estimate given that low light and clouds likely obscure events. About twice as many events were observed to occur annually between 2002 and 2012 (Georgsdóttir, 2012), and preliminary results for 2010 to 2016 show a similar increased frequency especially with increases in 2010 after the Eyjafjallajökull eruption due to subsequent volcanic ash resuspensions. The prolonged melting season on Icelandic glaciers is captured by SENTINEL-2 images, which show suspended dust as well as older ash layers in the glaciers, both of which affect the surface albedo (Figure 5). Although

MODIS observations correspond to the lowest dust frequencies relative to meteorological observations, daily images from MODIS captured high frequency dust events during warm autumns in north-eastern Iceland in 2015 and 2016. Annual dust deposition of 17 gm$^{-2}$ clearly affects albedo, which for example was observed to be reduced by 0.36 (albedo dropped from 0.86 to 0.5) after a Vatnajökull glacier dust event, leading to increasing snow melt of ~ 0.6 m annually (loop E→C in Figure 1; Wittmann et al., 2017).

In addition to increasing frequency Icelandic dust storms exhibit spatial and temporal variability. Storms are most frequent in the north in summer, while in the south the dust storms are more frequent in late winter. Additionally, there is significant inter-annual and decadal dust storm variability (Dagsson-Waldhauserova et al., 2013, 2014a). With a combination of *in situ* observations, numerical modelling (HIRLAM model with 5 km horizontal resolution), and meteorological time series analysis, Arnalds et al. (2014) have assessed individual dust storms to estimate the quantity

of annual Icelandic dust emissions. They estimate an average of 31-40 Tg of dust is suspended annually, with 12-35% of that reaching the ocean before being redeposited (Arnalds et al., 2014; see Figure 4). Further numerical simulations of wind during dust storms has revealed the local nature of the windstorms, underlining the need for high atmospheric and topographic spatial resolutions for accurate simulations (Dagsson-Waldhauserova et al., 2016; Baddock et al., 2017). In another study using FLEXDUST and FLEXPART simulations, and meteorological re-analysis data Groot

Zwaaftink et al. (2017) estimated annual Icelandic dust emissions to be an order of magnitude lower at 4.3 ± 0.8 Tg with about 7% of that reaching the high Arctic (> 80° N). However, Groot Zwaaftink et al. (2017) likely underestimate dust activity from the hyper-active dust hot spots which produce high dust outputs, for example redistribution of material after volcanic events and glacial outbursts. Dust uptake processes are also poorly understood and Arnalds et



al. (2016) calculated deposition rates, which also included large particles with short travel distances < 100 km. That said both estimates illustrate the importance of high latitude dust production in the Arctic and the global context.

### 4.1.2 Biogenic SOA formation and its role for the growth of new particles into CCN over the sub-Arctic regions (loop B→E→C in Fig. 1)

The Arctic Ocean is surrounded by vast regions of tundra and Boreal forest, which during the summer period emit large quantities of BVOC. In order to fully understand how warming-enhanced BVOC emissions will influence the BVOC-aerosol-cloud-climate feedback loop in the Arctic climate system in the future, a detailed understanding of the processes governing the formation and growth of new particles is crucial. Within CRAICC the Lagrangian model ADCHEM was used to simulate the formation and growth of new particles along air mass trajectories originating from the Arctic Ocean

and traveling over tundra and Boreal forests before arriving at the Pallas Atmosphere–Ecosystem Supersite (67.97° N, 24.12° E; 565 m a.s.l.) in Northern Finland. A main focus was to constrain the processes governing the growth of new particles into the climatically important cloud condensation nuclei size range. Model simulations together with measured particle number size distributions reveal that during observed new particle formation (NPF) event days in Pallas, new particle formation begins when the air masses move from the Arctic Ocean inland over the Boreal forest.

The analysed event days are characterized by lower concentrations of particles larger than 50 nm in diameter ($N_{50\,nm}$), with median $N_{50\,nm}$ of 140 cm$^{-3}$.

According to the model simulations the particle growth can largely be attributed to condensation of low-volatility and highly oxygenated organic compounds (Figure 6; Öström et al., 2017). With the HOM condensation mechanism included ADCHEM can capture the initial growth between 1.5 nm and 20 nm in diameter during the observed new

particle formation events. However, the model still underestimates the particle growth between 20 and 80 nm in diameter and therefore underestimates the concentrations of particles in the CCN size range (diameter > 50 nm) the day following a nucleation event. Both in the model and observations the $N_{50nm}$ peaks at around 06:00 the day after new particle formation events with median $N_{50nm}$ of 1109 cm$^{-3}$ and 1674 cm$^{-3}$, respectively (Öström et al., 2017). The more than one order of magnitude increase in the $N_{50nm}$ concentration during the morning after NPF event days, in the model

and observations, indicates that growth from biogenic secondary material is an important process for maintaining CCN concentrations over sub-Arctic forests, as long as primary particle emissions are low.

Kyrö et al. (2014) studied the effect of reduced sulphur emissions from the Kola Peninsula on the aerosol growth, concentrations, and long-term NPF trends at SMEAR I in eastern Lapland, Finland. The frequency of NPF days (clear, class I events) was found to decrease by about 10% per year concurrently with decreasing SO$_2$ (also ~10% per year).

High concentrations of SO$_2$ and H$_2$SO$_4$ were found to promote the onset of nucleation by several hours and even catalyse some events without sunlight. In general, air masses coming over Kola Peninsula were found to favour NPF, with sulfuric acid explaining up to 50% of the observed growth of new particles. Decreasing sulphur emissions decreased the condensation sink (CS) by about 8% per year and thus increased the event-time $J_3$ (particle formation rate at 3 nm



particle diameter) by almost 30% per year. The biggest decrease in NPF frequency was during spring and autumn, when the importance of $H_2SO_4$ for growth was maximum and $H_2SO_4$ concentrations were too low to grow the particles to potential CCN sizes. Simultaneously, during these seasons surface temperatures are on average too low to compensate and enhance SOA formation. Collectively these effects have resulted in a 3-4 % per year decrease in the

potential CCN concentrations.

### 4.1.3 SOA in Arctic regions (loops D→A and E→A in Fig. 1)

Secondary organic aerosol sources and processes can be tracked using molecular tracers such as carboxylic acids and organosulphates. Carboxylic acids are typically of either biogenic or anthropogenic origin, while organosulphates are formed from reactive uptake of organics on acidic sulphate aerosols (Surratt et al., 2007; Iinuma et al., 2007). Formation

of organosulphates thus represents a mechanism for anthropogenic enhancement of biogenic SOA (Hoyle et al., 2011). Within CRAICC organosulphates were discovered in the Nordic and Arctic environment (Kristensen and Glasius, 2011; Hansen et al., 2014; Nguyen et al., 2014a; Kristensen et al., 2016). Hansen et al. (2014) measured the first year-round time searies of Arctic organosulphates and identified elevated levels during the late winter haze period at both the Villum Research station and in Svalbard, probably due to the influence of anthropogenic sulphate aerosols. Another

recent study in Northern Europe during winter points to coal combustion (i.e. high-sulphur fuel) as an important precursor of organosulphates in aerosols (Glasius et al., 2018).

The first annual time series measurements of humic-like substances (HULIS) in Arctic aerosols were also obtained within CRAICC (Nguyen et al., 2014b). HULIS consists of complex, relatively high molecular weight polyacidic organic molecules (e.g. Decesari et al., 2000) and influences the light-absorbing properties of aerosols. The average

concentration of HULIS was measured to be higher during months with less sunlight (November–April, 11 ng C m$^{-3}$) than during the remainder of the year (4 ng C m$^{-3}$) (Nguyen et al., 2014b).

A prominent feature of organic aerosols is their aqueous surface activity. Already two decades ago, Facchini and co-workers (1999) recognized the large potential impact of surface tension on aerosol-cloud interactions, but the subject has remained a source of recurring debate in the aerosol chemistry and cloud microphysics communities (e.g. Ruehl et

al., 2016; Ovadnevaite et al., 2017). Building on the work of Sorjamaa et al. (2004) and Prisle et al. (2008 and 2010), researchers in CRAICC studied hygroscopic growth and cloud forming potential of laboratory synthesized organosulphates (Hansen et al., 2015) and commercial model-HULIS samples (Kristensen et al., 2014) in laboratory settings, using HTDMA and static thermal gradient or continuous flow CCNC chambers. These studies demonstrated the presence of complex surface tension effects, surface partitioning, and salting out, and significantly broadened the

understanding of surface activity for water uptake and cloud activation of atmospheric organics with very different origins and chemical characters.



### 4.1.4 Ice nucleation

Cloud processes can also be influenced by ice nucleation and ice nucleating particles (INP), those particles which assist in heterogeneous nucleation and growth of atmospheric ice (cf., Vali, 2015). Within CRAICC a combination of particle and surface measurements have been utilized to compare and contrast how quantifiable material and thermo-kinetic properties affect ice nucleation efficiency in relation to the thermodynamic driving force. Field measurement campaigns have been initiated in an effort to quantify ice nucleating capacity of Arctic and Nordic air masses, while simultaneously idealized laboratory experiments have been conducted to examine the underlying physicochemistry of heterogeneous nucleation.

### 4.1.4.1 Laboratory studies

Idealized molecular beam and light scattering experiments have focused on building a fundamental understanding of how critical scale ice clusters initially form and grow vis-à-vis vapor deposition on well-characterized surfaces (Kong et al. 2012, Thomson et al. 2015). Observations demonstrate that surfactants play an important role due to their ability to enhance and/or suppress the adsorption and desorption kinetics of atmospheric particles (Kong et al. 2014a, Papagiannakopoulos et al. 2014, Lejonthun et al. 2014, Thomson et al. 2013, Johansson et al. 2017). In particular organic hydrocarbon surfactant layers - that for experimental purposes are used to model atmospherically relevant organic layers - have size and temperature dependent effects on ice nucleation, growth morphology and molecular water uptake (Kong et al. 2014b, Papagiannakopoulos et al. 2013). An important implication of such findings is that common atmospheric surfactants, such as volatile organic compounds can promote and/or hinder water uptake, changing the hydrophilicity of atmospheric nanoparticles. Additional, laboratory investigations focused on how soluble salts may nucleate ice and whether or not at below eutectic temperatures sea salts might act as ice nucleating particles rather than deliquesce as CCN (Kong et al. 2018, Castarède & Thomson 2018). All of these processes may affect cloud evolution and lifetime and thereby impact fundamental environmental processes like water cycle and radiative balance. These effects may be more pronounced in Polar Regions where the sensitive balance between liquid and solid cloud particles and their thermodynamically unstable coexistence in mixed phase clouds may also be influenced by environmental feedbacks.

### 4.1.4.2 Field studies

The INP counting and characterization instrumentation (Section 3.2) introduced to the Nordic region through the CRAICC collaboration was used in various field studies. The PEAC7 electrostatic deposition collectors and subsequent FRIDGE analysis (Schrod et al. 2016) were used to initiate the acquisition of building time series data for INP concentration in the Arctic environment, where very little observational data currently exists. A two-year time-series of measurements was established in Svalbard in conjunction with a global data series initiated through the EU FP-7



BACCHUS project (data currently being analysed). In an additional study the PEAC7 was used to sample sea-faring ship emission plumes transiting the Port of Gothenburg, Sweden. In those measurements, an amplification of ice nucleating capacity was observed from ship emissions (Thomson et al. 2018). As such the observed amplification may be important to consider as shipping routes open in Arctic waters as sea-ice loss increases (Possner et al. 2015, Peters et al. 2011, Fuglestvedt et al. 2014).

In a very recent field campaign – HyICE -- aimed at investigations of ice nucleation within the boreal environment the PINCii instrument developed within CRAICC was first field-deployed. In the context of HyICE data for instrument intercomparison was also collected using multiple existing INP measurement platforms (PINC, FRIDGE, SPIN, et al.). Although data analysis is ongoing, strong INP signals in agreement with other instruments and the water droplet breakthrough limit were clearly observed during the campaign (Figure 7).

### 4.1.5 Boundary layer stability in the Arctic (loop B→E in Fig. 1)

The boundary layer in the Arctic differs in a number of ways from its counterpart in the mid-latitudes. Away from the Arctic (and Antarctic) the diurnal variation of the surface heating controls the dynamics of the boundary layer. During daytime, atmospheric mixing is driven by the sun heating the surface and wind shear throughout the boundary layer (Batchvarova and Gryning, 1991). After sunset, turbulence decays and a new neutral and stable layer with little turbulence forms over the ground. In the Arctic, the diurnal variability is absent throughout most of the year. During the long dark winter period, a long-lived shallow stable boundary layer is expected to form. The layer is shallow partly due to surface cooling during the Arctic night and partly due to the large Coriolis force at high latitudes. If resilient enough the stable layer may even extend through the Arctic spring/summer after the sun is continuously above the horizon. Such conditions are favourable for the creation of atmospheric waves and low-level jets, both of which influence the atmospheric boundary layer and atmospheric exchange processes. However, very little is known about these processes and their importance for the climate system.

Convective boundary layers driven by upward heat flux due to ground warming are also expected in the Arctic. Even in inner Antarctica at the Dome C research station, which is characterized by very low temperatures, Argentini et al. (2005) observed convective boundary layers. The formation of the Dome C convective boundary layers is likely connected to the very low ambient temperatures – typically -30°C. King et al. (2006) argue that at such low temperatures the boundary layer formation is due to partitioning of available energy into sensible heat flux.

Figure 8 shows an example of the boundary layer height evolution with time, over two weeks during a period of continuous Arctic sun at the Villum Research Station in the high Arctic. The land surface is fully covered by snow during the observations and the sun is visible throughout the entire day with a 13° elevation angle that only varies diurnally by about 1°. The diurnal cycle that is characteristic for the behaviour of the boundary layer at mid-latitudes is absent. Instead, the boundary layer depth remains ~250 meters over the entire 5-day period with occasional outbreaks of wave like motions. Thus, it is clearly not representative of an idealized boundary layer but is rather influenced by





effects that are not yet fully understood. Figure 9 illustrates the conditions at the same location in mid-July. For the selected days the boundary layer is about 100 m and again considerable wave like motions are observed.

**4.2 Cryosphere**

**4.2.1 BC in snow - field measurements (component C in Fig. 1)**

In Sodankylä, Finland (67°22' N, 26°39' E) BC and OC in snow were measured in connection with broadband and spectral snow albedo, and other ancillary parameters of seasonal European Arctic snow. Measured low UV albedo values were partly explained by the properties of melting snow and also UV absorption caused by impurities in the snow (Meinander et al. 2013). The results revealed that the OC in the snow could be an important source of absorption. The OC absorption may also partly explain the high mass absorption cross section (MAC) value needed for BC to

match the measured low albedo values with the SNICAR (Flanner et al. 2007) simulated albedo. To derive optical properties for OC that is considered representative of the OC in the snow, spectrally resolved MAC estimates for the target are needed.

The SoS field experiment showed that soot has a negative effect on snow albedo (Svensson et al., 2016), however, the measured effect was not as pronounced in outdoor conditions as in a previous laboratory study (Hadley and Kirchstetter,

2012). In Figure 10, the albedo of the snow from the SoS experiments first three days and the corresponding EC concentration is presented, where the figure also contains a fit to the experimental data. In the experiments, it was observed that the absorbing contaminants on snow enhanced the metamorphism of snow under strong sunlight conditions (Peltoniemi et al., 2015). Immediately after deposition, the surface of the contaminated snow pack appeared darker than the natural snow in all-viewing directions, but the absorbing particles sank deep into the snow within

minutes. The nadir measurement remained the darkest, but at larger zenith angles the surface of the contaminated snow was almost as white as clean snow. Thus, for a ground observer the darkening caused by impurities can be completely invisible, causing overestimations of the albedo, while a nadir-observing satellite sees a more absorbing surface, thus underestimating the albedo (Peltoniemi et al., 2015).

Atmospheric deposition not only influences albedo but also other physical properties of snow and ice (e.g., Meinander

et al., 2014). Experiments focused on physical characteristics of snowpack utilized artificially added light-absorbing impurities, which decreased the density of seasonally melting natural snow (Meinander et al. 2014). The data suggests that BC can decrease the liquid-water retention capacity of melting snow. No relationship was found in case of natural non-melting snow. The significance of these results comes via the fact that snow density multiplied by snow depth equals the important climate model parameter of snow water equivalent (SWE).



### 4.2.2 Dust-, BC-, OC- and volcanic ash-snow field campaigns (loop C→A→B in Fig. 1)

Dust aerosols interact with rain, snow and ice during dust storms. Suspended dust was observed during wet and low wind conditions (Dagsson-Waldhauserova et al., 2014b) and the phenomenon of Snow-Dust storms has been documented (Dagsson-Waldhauserova et al., 2015). Icelandic dust is therefore likely to affect the cryosphere in Iceland
and elsewhere (Meinander et al., 2016). We hypothesize that Icelandic dust on snow can be one of the causes for Arctic amplification. This effect is explained as a positive feedback loop where Icelandic dust deposited on snow decreases surface albedo and increases solar radiation absorption and snow and ice melt (Figure 11). Such effects are normally linked to soot but volcanic ash and dust from Iceland may actually have a larger influence in the Arctic region. This is in contrast to what is generally concluded on crustal dust climate effects in the latest IPCC report.

Iceland is the largest and most active high-latitude dust source, where dust deposition is expected to influence an area larger than 500,000 km$^2$ (Arnalds et al., 2014; Dagsson-Waldhauserova et al., 2014a). About 50 % of the annual dust events in the southern part of Iceland take place at sub-zero temperatures or in winter (Nov-Apr), when dust can be mixed with snow. This implies that the impacts and significance of Icelandic eruptions and dust events may be seriously underestimated. In studies on aerosols effects on Arctic climate the effect of mineral dust is often not explicitly
considered (e.g., Yang et al., 2014; Najafi et al., 2015) although paleoclimate data-archives indicate a substantial contribution of dust to radiative cooling in the Arctic (Lambert et al., 2013). Climate change can also increase extreme wind events and dust storms. Human initiated deforestation has also enhanced the desert area in Iceland, which coupled with changing climate can increase the probability for increasing dust event occurrences (Chapter 12 in Arnalds, 2015). Such positive feedbacks between snow and impurities were investigated during several outdoor and indoor experiments
with applications of Icelandic dust, volcanic ash and BC on snow surfaces in Lapland and Finland (Meinander et al., 2014, Dragosic et al., 2016, Peltoniemi et al., 2015, Svensson et al., 2016). The outdoor SoS 2013 experiment showed that volcanic dust on snow causes reductions in albedo and increases snow melting similarly to that of BC. Spectral reflectance of Icelandic dust was observed to be as black as soot and measured at 400-2500 nm to remained < 0.04 which is typical for highly absorbing particles (Peltoniemi et al., 2015). In fact, the snow albedo effect of light absorbing
aerosols and the use of satellites to measure the darkening of the Greenland ice sheet (GrIS) has been the subject of a recent controversy in the scientific literature (Polashenski et al., 2015; Tedesco et al., 2016). Polashenski et al. (2015) studied satellite-observed trends in the Greenland ice sheet albedo and found no evidence supporting a hypothesis that these are caused by deposition of BC or dust. Instead, they attribute the albedo trend observed by MODIS Terra (but not by MODIS Aqua) to uncorrected sensor degradation. Tedesco et al. (2016) in turn explain that Polashenski et al.
(2015) use a daily product, which has been shown to deteriorate with latitude. Instead, Tedesco et al. (2016) use the GLASS surface albedo product, a 16-day integrated product which is a combination of data from MODIS and AVHRR and accounts for directional (BRDF) effects at high solar zenith angles, and observed a statistically significant decrease of the surface albedo over the GrIS. They compare their observations with model results and conclude that discrepancies

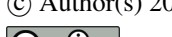


are due to the absence of light absorbing impurities in the model. Furthermore, the observed albedo trend is confined to regions of the GrIS that undergo melting in the summer with the dry-snow zone showing no trend. Both papers point at the absence of trends in the concentrations and deposition of either BC or dust, which leads to the conclusion that the observed albedo trends are not caused by changes in deposition of these species. However, rather than ascribing the

trend to sensor degradation, as in Polashenki et al. (2015), Tedesco et al. (2016) suggest that the albedo trend is caused by the exposure of a "dark band" of dirty ice and to increased consolidation of light absorbing material at the surface from melting. Within CRAICC, Meinander et al. (2016), addressing these concerns, concluded that an assessment of the effect of Icelandic dust on snow/ice surface darkening and melt is currently unavailable and therefore scientific research is critically needed. These authors hypothesize that, in the Arctic, Icelandic dust may have a comparable or

even larger effect on the cryosphere than soot (Bond et al., 2013). Observations and modeling results on Icelandic dust and cryosphere interactions for the past, present, and future are urgently needed.

Ash from the 2010 eruption of Eyjafjallajökull was used to find an insulating threshold for snow and/or ice. If the layer thickness of ash on snow or ice is very thin, it has the potential to increase snowmelt, but for thicker layers the snowmelt can be decreased compared to clean ice, due to thermal isolation. In experiments by Dragosics et al. (2016) and

Wittmann et al. (2017) an ash layer thickness of 9-15 mm on top of snow demonstrated an insulating effect, whereas a layer of only 1 mm increased melting to a maximum.

### 4.2.3 Changes of springtime snow-covered surface albedo (component C in Fig. 1)

Atlaskina et al. (2015), using thirteen years (2000-2012) of satellite observations from the Moderate Resolution Imaging Radiometer (MODIS), have conducted a study on snow-covered land surface albedo during spring in different

geographical areas of the Northern Hemisphere. The study showed that in the territories where snow cover fraction did not change and remained 100% throughout the study period, albedo has changed by ± 0.2 units over the 12 year period. The effects of air temperature, summertime enhanced vegetation index and precipitation amount and frequency on the surface albedo were investigated in attempt to explain the observed changes. A clear effect of the air temperature was found when monthly average air temperature exceeded a value between -15 to -10 °C, depending on the region, which

is colder than seen in the laboratory experiments. According to the laboratory experiments (e.g. Aoki et al., 2003), snow albedo is stable at temperatures colder than -10 °C. When temperatures reach -5 °C and higher, snow albedo decreases. Laboratory experiments are widely used to describe snow properties in models, but CRAICC research showed that they might represent atmosphere-cryosphere interactions unrealistically. This finding indicates that positive snow-albedo feedback takes place even at the relatively low temperatures and when snow cover is complete. The results indicate

that some Arctic areas are more sensitive to warming, and therefore can exhibit climate change at a faster rate. In the study, the relation between surface albedo and the other parameters was less clear in comparison to temperature effect or even absent in most regions.



### 4.3 Vegetation and land-use

#### 4.3.1 Vegetation-climate interactions in the past (loop A→B→C in Fig. 1)

When European Arctic treeline LPJ-GUESS model simulations are compared with the locations of modern treeline detected from satellite imagery and past treeline determined using proxy-based reconstructions, the simulated treeline agrees with the spatial distribution of the actual treeline when a threshold biomass value of 2 C kg m$^{-2}$ is used (Fang et al., 2013). Mismatches are primarily observed over mountainous regions, such as in northernmost Fennoscandia and in regions near the Ural Mountains. The likely cause is that the spatially distributed climate data used to drive the climate model are limited in terms of representing the varying climate conditions between valleys and mountain peaks. Inaccuracies and simplifications in species-specific simulations are also caused by the parameterization of coniferous species. For example, one significant factor contributing to low simulated biomass of pine in Fennoscandia modern treeline vegetation is that in the LPG-GUESS simulation pine is classified as a shade-intolerant species and is therefore suppressed by shade-tolerant spruce when the ranges of these two species overlap. Such comparisons suggest needed improvments include the spatial resolution of climate data within regions with complex terrain, such as steep slope gradients, and vegetation models need to simulate mixed forests composed of species with close bioclimatic thresholds.

Due to the inherent difficulties of species-specific simulations the CRAICC focus was on simulating the dynamics of coniferous treeline instead of species-level treeline shifts. The simulations and proxy data both indicate a northward expansion of treeline in the middle Holocene and a retreat during the late Holocene. In Figure 12 northward expansion towards the current tundra is indicated by the rise of spruce pollen values at 9 ka, and the late-Holocene retreat by the decline of spruce pollen values after 5 ka. When the simulated treeline dynamics are compared with these data, it can be seen that the model correctly captures the northward expansion of the boreal forest during the mid-Holocene and simulates realistically a treeline retreat in response to climate cooling during the last 3000 years. However, there are data–simulation disagreements particularly during the early Holocene, which mainly result from the differences between the two paleoclimate model scenarios used to drive the simulations. Thus, the results show that the LPJ-GUESS dynamic vegetation model does not perform particularly well for species-specific biomass simulations. However, the general advances and retreats of the arctic treeline can be realistically simulated with the dynamics vegetation model. In the model, the northward advances of the treeline are positively related to temperature variations, leading to a positive feedback loop between temperature and treeline.

#### 4.3.2 Warming-enhanced biogenic emissions from boreal forest (loop A→B→E in Fig. 1)

Boreal forest landscapes, including the boreal soils, waters, and vegetation are well known sources of biogenic emissions (Penuelas et al. 2014). A CRAICC modelling platform was applied to assess the potential Earth system feedback linking BVOC emissions from boreal forests, and the resulting SOA formation, to the direct and indirect aerosols and climate effects. Two experiments were performed with NorESM1-CRAICC to quantify the effects of



BVOC feedbacks, using a slab-ocean model and the Community Land Model (CLM) with interactive BVOC emissions calculated according to the MEGAN module. While including the BVOC feedback mechanism, the model was run for a pre-industrial $CO_2$ level of 357 ppm (*1xCO$_2$-FB*) and a $CO_2$ doubling (*2xCO$_2$-FB*) scenario. To achieve equilibrium the model was integrated for 70 years. In the boreal forest region, monoterpene emissions increased from 18 Tg yr$^{-1}$ to

28 Tg yr$^{-1}$ due to climate change from doubled $CO_2$. The increase in monoterpenes led to nearly doubling the simulated SOA formation and a 10% increase in total particulate organic matter (POM). NorESM1-CRAICC simulates the effect of SOA formation on 1) nucleation, 2) nuclei growth and 3) bulk SOA mass formation (Makkonen et al., 2014). The effect of the 10% increase in POM can be attributed to the changes in aerosol size distributions resulting from several competing pathways. Increased POM contributes to increased coagulation and is a condensation sink, possibly

decreasing nucleation and subsequent growth. Furthermore, the aerosol size distribution is modified by simulated changes in cloudiness and precipitation. Indeed, cloud cover over boreal forest increases from 54.1 % to 56.9 % and precipitation intensity increases slightly from 0.09 to 0.10 mm h$^{-1}$.

NorESM1-CRAICC was also used to simulate an alternative climate-warming scenario, in which biogenic VOC emissions were not allowed to change ("no feedback" experiment, 2xCO$_2$-NOFB). Thus, the two simulations, one with

and one without BVOC-aerosol-climate feedback, allow us to quantify the BVOC effect on aerosols and climate. In a control (1xCO$_2$) simulation, total particle concentrations over boreal forest regions averaged 820 particles cm$^{-3}$, while in the climate change simulation without BVOC-aerosol feedback showed particle concentrations decreased to 790 particles cm$^{-3}$, but the increased SOA formation in 2xCO$_2$-FB simulation resulted in increased concentrations of 880 particles cm$^{-3}$. Similarly, the vertically integrated cloud droplet number concentrations increased from 1.9x10$^6$ cm$^{-2}$ to

2.6x10$^6$ cm$^{-2}$ with BVOC-aerosol feedback and only to 2.3x10$^6$ cm$^{-2}$ without feedback, when simulating a doubled $CO_2$ scenario.

The strength of the simulated BVOC-aerosol-climate feedback is clearly weaker over tropical regions compared to boreal forest. In tropics, monoterpene emissions are shown to increase 20% resulting in POM increase of only 2%, due to simulated climate change. However, even at the global scale, BVOC-aerosol-climate feedback can increase SOA

formation by 45% in a doubled $CO_2$ experiment, showing the potential for a strong feedback mechanism during the 21st century.

### 4.3.3 Aerosol-climate effects of land-use change (loop D→E→A in Fig. 1)

The CRAICC modelling platform was also applied to assess the importance of land-use perturbations for atmospheric chemistry and aerosols. While the albedo and GHG effects of land-use are widely recognized, an accounting of the

changing spectra of primary aerosol and BVOC emissions is largely missing. Anthropogenic land-use change at high-latitudes could play a role in modifying aerosol-climate feedbacks through direct and indirect aerosol effects. A total of four simulations (Table 1) were performed using NorESM1-CRAICC to study the effects of land-use change on BVOC emissions, atmospheric chemistry and SOA formation.



Since 1850 there has been significant tropical deforestation in South America, South Asia and Africa, and considerable extra-tropical deforestation of mixed and boreal forests in Eurasia and North America. NorESM1-CRAICC shows large reductions of BVOC fluxes over these regions as a result of deforestation. Globally, when isoprene and monoterpene emissions are modelled and all other factors controlling BVOC emissions are kept fixed, the emissions are modelled

to be about 10% lower in 2000 than in 1850 due to land-use change (comparing simulations 1 and 2 in Table 2). The global reduction of BVOC emissions leads to an increase in OH concentrations in the boundary layer (BL) of up to 40% over regions marked by strong deforestation because BVOCs are important compounds controlling OH reactivity and abundance through direct reactions. Simulations 1 and 2 also show that BL ozone concentrations are reduced (up to 10%), because BVOCs serve as ozone precursors, when $NO_x$ concentrations are high.

Simulations 3 and 4, indicate that lower BVOC emissions lead to a reduction in SOA formation, lower loadings of organic aerosol (Figure 13a) and lower cloud droplet concentrations (Figure 13b) if oxidants are fixed at present-day levels. The effects of deforestation are seen as decreased organic aerosol burden and cloud droplet number concentration. The changes in organic aerosol are particularly marked in the northern hemisphere including large portions of Boreal forests in Eurasia and North America.

**4.4 Ocean and lakes**

**4.4.1 Arctic sea ice**

The thin veneer of sea ice which covers much of the Arctic Ocean is an important variable for the energy and ecosystem balances of northern latitudes. A historical minimum in sea-ice extent occurred in 2012, during which time CRAICC joined the CHINARE Arctic expedition to carry out various observations on sea ice and other atmospheric and

oceanographic conditions (Lei et al., 2014, 2015). Additional satellite imagery analysis using data from 1979 to 2012 to analyse interannual, seasonal, and spatial changes in the sea ice, shows that general thinning of the ice cover and delayed freeze up result in high variability during the month of October (Lei et al. 2015). Changes to the freezing and melting cycles of that ice are also important for the survival of multiyear sea ice (e.g., Rothrock and Maykut, 1999; Wadhams, 2016; Johannessen et al., 2017). Seasonal ice forms annually due to sub-freezing winter temperatures, while

multiyear sea ice is connected to summer warming because the equilibrium thickness is highly sensitive to summer melt. Landfast ice around the Arctic Ocean coasts and islands is seasonal and its mechanics are sensitive to the ice thickness and forcing by winds and tides (Yang et al., 2015). Sea-ice transport out of the central Arctic ocean is a significant sink for ice that is sensitive to large scale atmospheric circulation, for example the Arctic Oscillation. Even just a few decades ago most of the central Arctic was covered by multiyear ice 3–4 meters thick. Now a larger and

larger fraction of the central Arctic has experienced ice–free periods, and the thickness of multiyear ice has decreased by one meter (Wadhams, 2016). In a future, warmer Arctic sea-ice extent and thickness may also experience qualitative



changes, for example more snow-ice and frazil ice may form as a result of different freezing pathways. Present Arctic sea-ice models do not well represent the physics and uncertainties of such future sea-ice cover scenarios.

### 4.4.1.1 Sea-ice cover modulating marine emissions (loop C→A in Fig. 1)

Changes in sea-ice cover and sea temperatures also affect marine emissions to the atmosphere including sea-salt
aerosols, organic sea spray, and dimethyl sulphide (DMS) (Struthers et al., 2011; Nilsson et al., 2001; Browse et al. 2014). There exist a number of secondary processes that are also likely to effect marine emissions under large scale melting. For example, bubbles are generated on melting sea ice (Norris et al., 2011) leading to enhanced sea spray formation because they protrude from the ocean surface (de Leeuw et al., 2011). The melting of sea ice also, at least temporarily, causes a brackish surface layer of water that has been observed to produce less, but bigger bubbles and
therefore less sea spray particles (Mårtensson et al., 2003). The issue of how the surface melts is also important. Sea spray particle flux measured over open leads, was found to be an order of magnitude smaller than measured over the open sea (Nilsson et al., 2001). Thus, while an overall trend of increasing marine emissions is expected with decreasing sea ice, the spatial geometry of the melting will be important in addition to the effects of changes in salinity, fetch, biological activity, etc. Taken together, these changes will alter the character of the Arctic atmospheric aerosol and thus
likely play a role in Arctic cloud formation and cloud processes.

Struthers et al. (2013) combined global climate model output with an emission parameterization to estimate the change in regional and global sea salt aerosol number emissions from 1870 to 2100. Globally averaged, a general increase in sea salt aerosol number emissions was found, due to increasing surface wind speed. However, emissions changes were not uniform over the aerosol size spectrum due to an increase in sea surface temperature. From 1870 to 2100 the
emissions of coarse mode particles increased by approximately 10% (global average), whereas no significant change in the emissions of ultrafine mode aerosols was found over the same period. Significant regional differences in the number emissions trends were also found. Based on global climate model output from CAM-Oslo (Seland et al., 2008; Kirkevåg et al., 2008), the predecessor of CAM4-Oslo in NorESM1, no straightforward relationship was found between the change in the number emissions and changes in the sea salt aerosol burden or optical thickness. This was attributed
to a change in the simulated residence time of the sea salt aerosol. For the 21st century, a decrease in the residence time leads to a weaker sea salt aerosol-climate feedback than what would be inferred based on changes in number emissions alone. It should be noted that the above cited simulations of sea spray in CAM4-Oslo and NorESM1, including Struthers et al. (2013), did not apply an improved temperature dependent sea spray source parameterization (Salter et al., 2015) built on data from a new Stockholm sea spray simulation tank (Salter et al., 2014). The Salter et al., (2015)
parameterization is likely to result in more temperature sensitive emissions compared to the previously used parameterization by Mårtensson et al. (2003).

Sea spray simulation chambers like the Stockholm simulation tank developed and/or deployed within CRAICC (Section 3.2.3) have enabled investigations of the physical and chemical properties of sea spray aerosol. The cloud forming



ability of sea spray aerosol has been targeted in a series of studies coupling CCN counters to sea spray tanks containing real and artificial sea water samples (REFs). King et al. (2012) established a baseline for artificial sea salt CCN activity and found that particle generation method (diffuser vs. plunging jet) can affect particle size distributions and particle properties. During two spring campaigns Rasmussen et al. (2017) investigated the CCN activity of particles generated

from sea water samples collected in the bay of Aarhus, DK. The measured CCN activity was similar to particles generated using artificial sea salt and did not vary significantly over the sampling period. Another laboratory study dedicated to the effect of saturated (solid) and unsaturated (liquid) fatty acids on sea spray aerosol properties showed that a significant organic volume fraction of saturated fatty acid (>60%) was needed to decrease CCN activity compared to that of a purely inorganic sea salt particle of the same size (Nguyen et al. 2017). Salter et al., (2016) presented

observations of the size-dependent enrichment of calcium in sea spray aerosol generated from both artificial seawater and natural seawater collected in the North Atlantic. A significant result from CRAICC has been to clarify the possible role of hydrates in dried sea salt particles on climate relevant properties. Rasmussen et al. (2017) used a sea spray simulation tank coupled with a thermodenuder to show that sea spray volatility can to a large extent be ascribed to the presence of hydrates, while Zieger et al., (2017) used the Stockholm apparatus and found it likely that to the presence

of hydrates reduced the hygroscopic growth of inorganic sea spray aerosol by 8-15% compared with pure sodium chloride.

Connecting fundamentals of sea spray from bubble formation and bursting to atmospheric aerosol processes is key to understanding the complex feedbacks associated with changing Arctic surface conditions, as enumerated in Figure 1, loop C→A.

### 20    4.4.1.2 Long Icelandic record of sea-ice extent (component C in Fig. 1)

Iceland represents an underutilized source for Arctic sea-ice recording. Meteorological observations have been carried out in Iceland since the early 1800s although the Icelandic Meteorological Office was not established until 1921. In place of scientific observations, regional historical sources indicate the extent of glaciers and the presence of sea ice, centuries back in time (Björnsson, 2016, Ogilvie and Jónsdóttir, 1997). For example, information on monthly sea-ice

extent goes back to 1850 based on farmers' diaries and ship captains' logbooks (Jónsdóttir, 1995). Systematic observations begin in the early 1900s (DMI, 2018) and are more recently complemented by the much higher temporal and spatial resolution obtained from the 1970s onward during the satellite era. Geologic indicators such as sediment records also provide insight into environmental change over the last millennia (Knudsen et al. 2009, Larsen et al. 2011). Sea-ice extent off the Icelandic coast shows considerable variation with time. Conditions were severe throughout the

second half of the 19th century and into the 1920s, with sea-ice blocking the north coast for weeks or months in most years, often with serious implications for fishing, farming and transport (Jónsdóttir, 1995). Much milder conditions followed and were observed up to 1965, when harsh conditions returned for more than a decade. The 21st century has




seen dramatic changes in Northern Hemisphere sea-ice cover with decreased ice extent, thickness and proportion of multiyear ice within the ice pack (NSIDC 2018, Wadhams, 2016, Lei et al., 2014, Jónsdóttir and Sveinbjörnsson, 2007).

**4.4.2 High-latitude lakes and lake-atmosphere interactions (loop B→C & E→A in Fig. 1)**

Most lakes on Earth are located at high latitudes in the Arctic and boreal regions where they are covered by ice during winter (Verpoorter et al. 2014). Changing cryospheric conditions are critical to how these lakes evolve and influence the global climate system. High latitude lake-atmosphere interactions are largely governed by surface temperature, waves, and the presence and character of ice cover. Lake processes can act on time scales of days to months, depending on lake size, while longer memory effects can extend over the length of the ice season. Lakes are active players in weather phenomena and budgets of gases, and large lakes and lake districts show up in regional climatology (Yang et al., 2013; Leppäranta, 2015). Numerous publications have shown that recent climate warming has affected lake icing, resulting in later freezing dates and earlier ice breakup dates, both on the order of 5–7 days, over the last century (Kirillin et al., 2012). In addition to higher air temperatures, break-up is forced by solar radiation which causes melting, throughout the ice column, where internal melting also contributes to mechanical weakening and ice cover breakage. Overall ice thickness may decrease due to milder autumn/winter conditions or less cloudy spring conditions that directly affect the radiation balance at lake surfaces. Trends in winter precipitation that effect snow amount can also affect ice thickness, and quality.

In contrast to land surfaces, turbulent heat transfer in lakes provides a strong smoothing mechanism for temperature variations. Surface roughness is less over lakes than over land and the water surface is continuously striving to achieve water vapor equilibrium vis-à-vis ongoing evaporation and sublimation, resulting in locally humid conditions. Ice cover has a major impact: turbulence is decayed, circulation becomes thermohaline, and the lake water body becomes decoupled from the atmosphere. Cold fresh water ice is largely non-permeable to gases but becomes permeable as it deteriorates during melting. Atmospheric deposition is stored within the ice cover during the ice season and released during the short melting period.

The annual cycle of boreal and tundra lakes can be divided into stratified summer conditions, the ice season, and mixing periods between them (Figure 14). The formation of the summer stratification necessitates that air temperatures rise to more than 4 °C, while freezing takes place only with sustained below freezing air temperatures and is delayed depending on lake depth. Lake ice response to climate warming is expected to manifest as reduced ice cover periods but also a changing of the quality of ice seasons. Early and late winter are characterized by periods when ice is weak and may be broken by wind. During the stable phase, ice cover is safe and thick enough to support on-ice traffic. In a warming climate, the stable period is expected to shorten due to the decreasing ice thickness, and eventually the entire ice season may become unstable.

Although lakes only comprise about 3.7% of the Earth's non-glaciated land area (Verpoorter et al. 2014) they are very efficient in emitting greenhouse gases. Recently, it has been shown that on a global scale as much as 2.1 PgC are



annually emitted from $CO_2$ reservoirs in lakes, ponds, and running waters (Raymond et al. 2013). This amount is comparable to the annual $CO_2$ uptake by oceans (~2.0 PgC yr-1; Wanninkhof et al. 2013). Thus, fresh water must also be considered as an important regulator within the global carbon cycle, and thereby also influence climate.

A large fraction of the $CO_2$ emitted from inland waters has its origin in surrounding terrestrial ecosystems. However,
substantial amounts are also produced by microbial and photochemical mineralization within the water column (Cole et al. 1994; Humborg et al. 2010; Weyhenmeyer et al. 2015). The production of $CO_2$ within the water column depends upon the availability of dissolved organic carbon (DOC), which enters inland waters from the surrounding watersheds. Kasurinen et al. (2016) developed models to quantify the DOC export from soils in northern watersheds. The modelling approach showed that DOC concentrations depend on watershed water storage as well as on soil temperature.
Consequently, changes in land use will have a major impact on DOC export and therefore $CO_2$ production in lakes. Each year during the cold winter months substantial amounts of $CO_2$ are produced and accumulated below lake ice cover (Denfeld et al. 2015; Denfeld et al. 2016a). Similarly, $CH_4$ is produced below ice cover, in particular when nutrients do not limit its production (Ricão Canelhas et al. 2016; Denfeld et al. 2016b). At ice break-up (already at melting stage) these gases can be emitted into the atmosphere. However, gas production in water below ice is still less
than production during the open-water season (Denfeld et al. 2016a). Thus, given sustained nutrient conditions, similar groundwater hydrological connectivity, and sustained biological $CO_2$ in the water column, it is also likely that gas production and emissions from inland waters will increase with longer and stronger cryosphere melt seasons (Denfeld et al. 2016a). It has been further suggested that these and other physical manifestations of global warming will impact ice phenology include time-delayed effects and feedbacks, which inspires continued research.

**4.5 Anthropogenic aerosols influence in the Arctic**

**4.5.1 Past BC deposition (loop D→A in Fig. 1)**

CRAICC has greatly increased the available BC deposition data from the Arctic. Previous observations of BC deposition were mainly from North American emission-influenced high-elevation Greenland sites (McConnell et al., 2007; McConnell and Edwards, 2008; McConnell, 2010; Keegan et al., 2014), which are not necessarily representative
of overall BC deposition trends at lower elevation Arctic locations. Many of those receive the majority of their BC loading from Eurasia (McConnell et al., 2007; Hirdman et al., 2010). The Greenland records show increasing BC deposition from 1850, peaking around 1910, followed by a decline to almost pre-industrial levels after 1950 (McConnell et al., 2007; McConnell and Edwards, 2008; McConnell, 2010; Keegan et al., 2014). CRAICC work includes pioneering determination of depositional BC fluxes from the atmosphere to a Svalbard ice core (Ruppel et al.,
2014) and four northern Finnish lakes sediments (Ruppel et al., 2015), challenging the prevailing conception of declining or stable Arctic BC values during the last decades.



The Svalbard and Greenland ice cores show similar temporal BC trends between 1750 and 1950, after which the Svalbard ice core indicates an increase in elemental carbon (EC, thermal-optical proxy for BC) deposition post-1970 (Figure 15A), deviating strongly from the Greenland records. This increase is hypothesized to be partly caused by increased flaring emissions from gas and oil extraction in northern Russia (Ruppel et al., 2014). Flaring has been a

previously under-estimated Arctic BC emission source but is suggested to account for 42 % of mean annual atmospheric surface BC concentrations in the Arctic (Stohl et al., 2013). These within-Arctic BC emissions do not reach the high-elevation Greenland ice core sites due to restricted isentropic uplift in the atmosphere between the emission sources and Greenland summit (Stohl 2006; Stohl et al., 2013), which helps to explain the differences between Greenland and Svalbard BC deposition trends.

Four northern Finland lake sediment records also show a similar standardized average soot BC (SBC) deposition trend in agreement with the Svalbard ice core with increasing deposition since 1970 (Figure 15B, Ruppel et al., 2015). Lake sediment BC records are more sensitive to external factors (e.g. delayed BC import from catchment areas) than ice cores, and thus only general trends observed in multiple records can be considered significant (Rose and Ruppel, 2015). The observed SBC deposition increase is likely caused by regional BC emissions from the Kola Peninsula, as BC

deposition from long-distance sources would have been expected to clearly affect all the studied sediment records (Ruppel et al., 2015). Thus, although the observed results cannot be extrapolated to wider regions, the fact that increasing BC deposition trends were observed in different environmental archives analysed with different methods suggests a robust result. Furthermore, the implication of the findings is that BC sources are considerable and that similar BC deposition trends may be observed in other parts of the Arctic.

Consequently, the CRAICC data suggests that BC deposition has increased since 1970 in some parts of the Arctic coinciding with an increasing climatic impact of BC. This highlights substantial gaps in the previous (AMAP, 2011, 2015; Bond et al., 2013) understanding of Arctic BC trends. Conclusions that Arctic warming has occurred during the last 20 years despite decreasing BC concentrations (AMAP, 2011) may have been premature. More observational BC deposition data are urgently needed to confirm the BC trends across other parts of the Arctic.

The Svalbard ice core and Finnish lake sediment BC records both underline the potential existence of strong BC emission sources within the Arctic (Ruppel et al., 2014, 2015). The presence of high-latitude BC sources is important because within-Arctic BC emissions have a higher probability to remain in the region and are known to result in an amplified Arctic surface warming (5x) compared to the same emissions originating at mid-latitudes (Sand et al., 2013a). If the BC deposition trend has increased since the 1970s over wider Arctic areas, BC may have hastened the retreat of

the Arctic sea ice, contrary to what has been previously thought (e.g., Doherty et al. 2010), based on temporal snap shots derived from BC in snow measurements, which show lower values between 2005 and 2009 than during the 1980s. SILAM chemical transport modelling results developed within CRAICC to investigate the Svalbard ice core data have the potential of increasing the general understanding of meteorological processes driving BC deposition trends. Modelled BC deposition trends at the Svalbard glacier show variations that appear independent from BC emissions or





trends in atmospheric BC concentration between 1980 and 2015. According to the model, about 99 % of BC mass is wet-deposited at the glacier, indicating that meteorological processes such as precipitation and scavenging efficiency have a stronger influence on the BC deposition than the emissions or trends in atmospheric concentration (Ruppel et al., 2017). Additionally, the model results indicate that, contrary to expectations, dry deposition does not follow a

similar trend to atmospheric concentrations across the Arctic, in particular at sites located closer to anthropogenic sources (Soares et al., in prep.). Thus, the model results do not point to a specific within-Arctic BC emission source being responsible for the observed post-1970s increase in European Arctic BC deposition, but highlight that temporal trends in atmospheric BC concentrations and deposition may diverge due to meteorological processes. Consequently, Arctic BC deposition trends should not directly be inferred based on atmospheric BC measurements, and more

observational BC deposition data are required to comprehensively assess the climate impact of BC in Arctic snow (Ruppel et al., 2017).

### 4.5.2 European emission reductions enhancing Arctic changes (loop D→A→B in Fig. 1)

With the reduction of atmospheric sulphate concentrations since the 1980s in Europe and North America the aerosol cooling effect has likely decreased. Using the CMIP5 version of the climate model NorESM1-M, Acosta Navarro et al.

(2016) found that the reduction of sulphate in Europe between 1980 and 2005 can explain as much as half of the warming observed in the Arctic during the same period. Air quality emission(s) regulations have succeeded in reducing soil and water acidification and have therefore also removed a large portion of the aerosol dampening effect and thus the Arctic is experiencing warming more in line with the increased global greenhouse gas levels. However, over continental Europe itself the modeled warming due to the same European emissions regulations is only one-fifth of the

observed warming (Kirkevåg et al., 2016). Acosta Navarro et al. (2016) found that a redistribution of the energy input to the Arctic over the year appears to be a critical factor in explaining the stronger temperature response in the Arctic. In short, the reduction of European anthropogenic sulphate emissions leads to increased summertime atmospheric heat transport (0.25 Wm$^{-2}$) and increased yearly oceanic heat transport into the Arctic region (+0.48 Wm$^{-2}$). Together with the increased ocean heat transport, the summertime energy surplus is mainly used to melt sea ice, thus increasing the

amount of available energy in the Arctic and delaying the onset of autumn freezing. Delayed freezing is reflected in a larger heat transfer from the ocean to the atmosphere during fall and winter causing strong lower tropospheric warming.

### 4.5.3 Contrasting mid- and high-latitude anthropogenic emissions (loop D→A→B in Fig. 1)

Most aerosol particulate reaches the Arctic through long-range transport. Arctic sites have measured surface concentrations of EBC ('equivalent' BC determined using light absorption methods) and sulphate for over three

decades. At these monitoring sites, northern Eurasia has been identified as the largest EBC source region (Sharma et al., 2004: Eleftheriadis et al., 2009; Hirdman et al., 2010), with contributions peaking in winter and spring. During



summer, BC emissions from agricultural and boreal fires have a primary influence on Arctic concentrations (Stohl et al., 2006, 2007). Within CRAICC Sand et al. (2015a) studied feedback effects of increasing BC in the atmosphere. The modeling experiment increased BC to unrealistically high levels to show that increased BC concentrations may change cloudiness and precipitation in ways that will feed back onto the spatial distribution and atmospheric residence times

of BC. In extreme cases, increased BC in the upper parts of the troposphere may contribute to stabilizing the atmospheric column below and destabilizing the column above, thus causing a positive feedback which mixes BC to higher altitudes and further from emission sources, e.g. to polar latitudes. These feedback effects are not accounted for in many model experiments wherein BC concentrations are directly controlled.

Using NorESM1 with an emission inventory that includes flaring and seasonal variations from domestic sources, Sand

et al (2013a) investigated how mid latitude BC emissions impact the Arctic in comparison with local BC emissions. They find that BC emitted within the Arctic has a five times higher surface temperature response in the Arctic compared to the same amount of BC emitted at mid-latitudes. A large fraction of this warming is due to increased absorption from BC deposited on snow/sea ice. BC emitted at mid-latitudes on the other hand, enters the Arctic at higher altitudes and is therefore less likely to be deposited on the surface. BC at higher altitudes may warm the surrounding layers, but the

high static stability of the Arctic atmosphere suppresses any heat exchange between the upper and lower atmosphere (Sand et al., 2013b). Motivated by the BC mitigation potential, Sand et al. (2015b) split the emissions into different sectors and regions, and provided detailed numbers for the surface temperature sensitivity in the Arctic per kg emission of BC, taking co-emitted species into account. They found that domestic emissions from Asia have the largest warming effect in the Arctic, due to the large absolute amount of these emissions and the relative high BC-to-$SO_2$ fraction.

However, the Arctic is most sensitive, per unit mass emitted, to Russian flaring emissions, followed by forest fires and flaring emissions in the Nordic Countries.

## 5. Legacy of CRAICC

The results presented throughout this review were only achieved by combining established and newly developed scientific methods and tools, and by combining these in new and innovative ways. While these individual scientific

results as such are indeed a major outcome of CRAICC, perhaps an even more important legacy is the way in which the new methods and tools were implemented in a synchronized manner in order to elucidate the feedback loops of interest (Figure 1). In doing so, they will further influence the polar science community and enhance our understanding of the complex Arctic climate system even beyond the CRAICC initiative.

The driving concept behind all research activities within CRAICC was to closely link and coordinate all efforts needed

to address our research questions related to the identified Arctic feedback loops. This was based on necessity, since no single research group had the capacity to undertake all the needed research. The combined expertise of the Nordic Earth system and Arctic climate science communities proved to be able to muster sufficient capacity to adopt the required holistic perspective on Arctic climate change.





As in all fields of natural science, we strived to integrate our combined knowledge into models – operating at different scales and levels of detail – in a condensed and operational fashion such that these models could be used to tackle important research questions. All models needed to be based on, and evaluated against, observations in order for them to gain credibility and acceptance. Such observations, in turn, were made both in the laboratory – where conditions can

be manipulated – as well as in field experiments and long-term monitoring efforts, which were typically subject to larger variability and less control.

CRAICC modelling platform: Modelling within CRAICC spanned a wide range of temporal and spatial scales, from molecular cluster dynamics models (Å and fs scales), to aerosol dynamics and atmospheric chemistry models (nm-km

and seconds-hours scales), to regional chemical transport models (km and h-days), to dynamic vegetation models, and finally to global climate and Earth system models (global and decadal scales). A major challenge – albeit not unique to CRAICC – had been to link models across scales. Rather than focusing on directly nesting models, by passing model domain boundary conditions between models while computing or off-line, the approach has been focused on passing scientific information and process understanding between model scales.

CRAICC experimental platform: The experimental platform in CRAICC also spanned scales and complexity, and involved laboratory process studies, campaign process and characterization studies, and long-term field observations across a geographically distributed network of stations. Designed laboratory and field experiments were used to derive and test parameterizations, specifically in an effort to distil descriptions of the complex web of interlinked processes

down to their dependence on controlling parameters. The level of detail allowed in the models, typically set by computing capacity, determined the complexity of the incorporated process parameterizations. All parameterizations needed to be justified and verified, which is what CRAICC was able to provide, owing to the wide-ranging expertise gathered in the research program.

Long-term field observation network: An extensive long-term field observation network extended from Sorø, Denmark to the Villum Research Station in North Greenland at 81°36' N. It built heavily on the existing environmental research infrastructure for aerosols, trace gases and clouds (ACTRIS, www.actris.eu), which has an ESFRI status as a strategically important research infrastructure in Europe. CRAICC also utilized observational data from the INTERACT Arctic station network, and gained access to some of the 79 INTERACT sites (http://www.eu-

interact.org/). Despite the existing extensive infrastructures at northern latitudes, there was an additional need for long-term observations in the pan-European Arctic, which led CRAICC partners to initiate the Pan-Eurasian Experiment (PEEX) Program (see PEEX section below).





Examples of CRAICC research "loop" to investigate feedbacks: Here we present two examples of how new methods, process studies and models were developed in a coordinated way within CRAICC, and how they were used to elucidate the feedback loops under scrutiny. Following the identification of important feedback loops involved in Arctic climate change, the proper combination of new scientific experiments, methods, tools and observations needed to examine the

feedbacks was designed and carried out within CRAICC. Whichever the feedback being studied, this followed a similar circular pattern, or "research loop":

Laboratory and field experiments → parameterizations for process and global climate models → model runs at various temporal and spatial scales → verification against field observations → feedback loops evaluated with the objective of rigorous quantification → identification of knowledge gaps and requirements for improved process understanding,

parameterizations, finer model resolution, better coupling between Earth system compartments etc. → new iteration.

The following are two examples of how this was carried out in practice within CRAICC.

Example 1: Warming-enhanced biogenic emissions from boreal forest (involved in loop D→A→B→E→A in Fig. 1)

The feedback loops could only be evaluated using models (alternative long-term data sets from SMEAR II, Hyytiälä,

Finland: T-BVOC-BSOA-AOD-CCN-higher albedo-cooling), given that this was the only way in which we could "experiment" with different scenarios.

*Laboratory and field experiments (smog chamber studies, SMEAR II etc.) → Parameterizations for process and global climate models (ADCHEM, NorESM1 etc.) → Model runs at various temporal and spatial scales (giving number concentrations fields) → Verification against field observations of CCN concentrations, OA etc. and their ageing time*

*scales and resulting atmospheric residence time → Feedback loops evaluated (leading to identification of knowledge gaps and requirements, for instance for improved process understanding, parameterizations, finer model resolution, better coupling between Earth system compartments etc.).*

Example 2: Sea spray number production (involved in loop D→A→B→C→D in Fig. 1)

Another key aspect of the work on sea spray aerosol conducted during CRAICC has been the tight links between the

laboratory experiments described previously and global modelling. The effects of temperature on sea spray aerosol production described by Salter et al., (2014) were implemented in NorESM1 as a sea spray source function (Salter et al. 2015), while Zieger et al., (2017) highlighted the implications of their measurements of inorganic sea spray aerosol hygroscopicity using ECHAM6-HAM2.

*Laboratory and field experiments (bubble chambers etc.: dependence on wind speed, bubble concentration and size*

*spectrum, sea water salinity and temperature, surface microlayer properties etc.) → Parameterizations for process and global climate models (ADCHEM, MATCH, CAM-Oslo etc.) → Model runs at various temporal and spatial scales (giving number concentrations fields) → Evaluation against field observations of sea spray number concentrations and their ageing time scales and resulting atmospheric residence time → Feedback loops evaluated (led to*





*identification of knowledge gaps and requirements, for instance for improved process understanding, parameterizations, finer model resolution, better coupling between Earth system compartments etc.).*

eSTICC (eScience tools for investigating climate change) is a Nordic Centre of Excellence that aims for a more accurate
description of the high-latitude feedback processes in the climate system by improving eScience tools (e.g., computational models, data platforms) for the climate research community. eSTICC originated mainly from the modelling activities in three Nordic Centres CRAICC, DEFROST (Impact of a changing cryosphere: depicting ecosystem-climate feedbacks from permafrost, snow and ice) and SVALI (stability and variations of Arctic sea ice). Nearly all of the eScience groups active in CRAICC have continued their cooperation in this new Centre. eSTICC has
pooled together researchers from 13 top institutes in the Nordic countries working in the fields of climate and/or eScience to improve eScience tools for climate research. Specifically, eSTICC develops tools needed for more efficient use of experimental and model data, and improve the computational efficiency and coding standards of Earth System Models (ESMs) and tools for inverse modelling of emission fluxes.

PEEX the Pan-Eurasian Experiment Program was initiated in year 2012 as a bottom up initiative by several European and Russian research communities (Lappalainen et al. 2014, Kulmala et al. 2015, Kulmala et al. 2016). It was an extension, not only, of the research approach carried out by the Finnish Centre of Excellence (Centre of Excellence in Atmospheric Science - from Molecular and Biological processes to The Global Climate) but also of the CRAICC initiative. In 2017 PEEX carried out the implementation of the PEEX Science Plan, which in many aspects shares the
same research interests regarding Arctic environments as those of the CRAICC initiative, but also expands the region of interest to cover the boreal environments of Northern Eurasian (Lappalainen et al. 2015). PEEX also actively involves socio-economic and social sciences in its research approach by linking socio-economic factors to systems analysis, scenarios, and narratives of the Northern future (Kulmala et al. 2016, Lappalainen et al. 2016). The understanding and controlling of GHG and SLCF emission(s) and concentration dynamics over the Russian Arctic-
boreal regions and the stability of Russian permafrost, currently occupying 65% of the territory (Melnikov et al. 2016), is one of the greatest scientific and social challenges for the entire northern Eurasian region. Thus, the environmental changes and energy policy choices taking place in Russia are especially relevant not only regionally but also globally, for the global climate-biosphere system.

Similar to one of the CRAICC's main interests PEEX aims at increasing the amount of available data from the Arctic-
boreal regions. This will be done by establishing a long-term, coordinated comprehensive *in situ* observation network across the Arctic-boreal regions of northern Eurasia (Hari et al. 2016). The concept of the hierarchical PEEX *in situ* station network is based on know-how of the measurement theory and techniques developed at the SMEAR (Stations Measuring the Ecosystem – Atmosphere Relations) -II flagship station in Hyytiälä Finland (Hari et al. 2016). The backbone of the PEEX station network is built on the existing biosphere (ecological) and atmospheric observation



networks in collaboration with European, Russian and Chinese partners. Starting from 2012 PEEX made preparatory work with the Russian station networks and collected preliminary information on *in situ* station measurements from over 170 stations (Alekseychik et al., 2016). The overview of the measurement capacity of the existing stations and analysis of future needs will the first step connecting the Russian RI and data formats more strongly to international

research infrastructure frameworks, especially to European research infrastructures like ACTRIS, ICOS, ANAEE and LTER. As a part of the PEEX infrastructure the "PEEX Modelling Platform" (MP) has been established. The MP combines multidisciplinary datasets of varying temporal and spatial scales and provides a seamless modelling interface for regional models and global earth system models (ESMs). The MP also provides different types of the visualization tools for comparing the measured dataset versus modelled data together with trajectory information. The MP also

enables connections between satellite observations and ground-based observations to promote a clear understanding of the current state of the atmosphere and for model validation.

Knowledge transfer: In the CRAICC community, we have recognized the importance of discipline-tied fundamental education for tackling multidisciplinary research problems. However, in climate and global change science a shift

towards multidisciplinarity is needed also in education (Nordic Climate Change Research, 2009). In CRAICC, we developed a model to improve the learning outcomes in multidisciplinary atmospheric science. The model included pedagogical experiments, utilization of modern technologies (e.g. Junninen et al., 2009), workshops for teachers and supervisors, and most importantly organizing a series of interdisciplinary research-intensive short courses. CRAICC had a dedicated education programs including master and doctoral level studies. On the master level, a new Arctic

study module was introduced in the existing joint Nordic master's program in Atmosphere-Biosphere Studies (ABS). On the doctoral level, the emphasis was on joint intensive courses, doctoral student mobility, and cross-supervision between CRAICC partners.

The total number of jointly organized intensive CRAICC research courses given was 21, and the total number of students participating in these courses was 484. The courses were interdisciplinary, and emphasized the following

aspects:

- Measurement techniques and field experimentation (6 courses)
- Advanced data analysis of atmospheric and ecosystem observations (7 courses)
- Theoretical approaches and basic understanding (4 courses)
- Arctic environment (3 courses)
- Modelling (1 course)

A total of 18 doctors were recruited to carry out research related to the topics studied in CRAICC. All of them graduated within one year after the project ended.



## 6. Summary and Outlook

The major CRAICC project aim was to quantitatively evaluate the identified feedback loops within the Arctic climate system with respect to changing climate and anthropogenic influences. While the most well quantified feedback loop in the Arctic climate system is the amplification of warming caused by the albedo changes with decreasing sea-ice extent, there remain significant areas of scientific uncertainty with regard to other Arctic system feedbacks and aerosol-cloud-climate interactions. For example, the changes in emissions of particulate and gaseous compounds due to rapidly decreasing Arctic sea-ice coverage will also have feedback effects on albedo (Stendel et al., 2008). Earth system modelling efforts carried out within CRAICC have contributed to both the identification and quantification of Arctic feedbacks. However, quantification of single feedback mechanisms remains difficult and the impact of further warming on ecosystems demands continued investigation.

The uncovering of Arctic waters was a major area of emphasis within CRAICC. Decreasing sea-ice extent will result in changes in the natural emissions (e.g. sea spray) from open water, it will also increase humidity levels, influence the formation of clouds, and the chemistry of the atmosphere. All of these will have consequential climate feedbacks. In particular DMS, amines, bromine and iodine containing organic and inorganic species will be fundamentally impacted. The increase in sea spray emissions from more open ocean and fetch will change their potential CCN contribution (potentially a negative feedback). Experiments with interactive DMS in NorESM1 have been done (Schwinger et al., 2017) indicating potentially strong regional Earth System feedbacks via DMS, and the mechanisms will be studied further in new model versions. However, in the long term the disappearance of sea ice ends the surface water temperature buffering near the melting point, which will allow the sea surface water temperature to gradually increase. For sea spray, that might reverse the emission trend. With increasing sea surface temperature, Mårtensson et al. (2003), Zábori et al. (2012) and Salter et al. (2014 and 2015) have shown that the sea spray emissions will decrease, resulting in a positive feedback on climate change.

The opening of the Arctic seas during summer will also make commercial shipping possible, but emissions from Arctic shipping have to be evaluated with respect to cloud and climate impacts in addition to the fuel and technological changes that might also evolve in future scenarios. Activity will be highly dependent on the Arctic sea-ice extent, which has been rapidly changing in the last decades. The emissions from ship transport, depending on the fuel, season, and cloud type, etc. may lead to warming and/or cooling effects. That said, shorter shipping routes will potentially lead to less global emissions.

Exchange processes between the atmosphere and the cryosphere are another area where CRAICC has contributed to narrowing the existing knowledge gaps. Although, similar processes govern the deposition of particles in the Arctic and mid-latitudes, particle emissions are different and can respond differently to climate change. Future emissions projections of primary particles and secondary aerosol precursors in a changing Arctic climate are handicapped because even present emissions are not well quantified. Furthermore, chemistry in the snowpack is not well understood and



BVOCs from thawing permafrost are expected to comprise a large and increasing fraction of Arctic and subarctic secondary aerosol precursor emissions. The oxidation and conversion of BVOCs to condensable organic compounds requires more scientific investigation, because the impact of secondary organic aerosols on e.g. cloud formation is still too uncertain to make reliable forecasts of Arctic climate, especially in a warmer Arctic. Further warming of the Arctic

will enhance BVOC emissions, which in turn will have significant impact on the organic aerosol mass load. Thus, the role of organic aerosol in the Arctic, including the role of organosulphates remains a key area for knowledge advancement (Hansen et al., 2014a and 2014b; Kristensen and Glasius, 2011; Öström et al., 2017).

A reduction in uncertainty of projections on BVOC emissions arising from both terrestrial and marine sources is needed in order to predict future Arctic climate scenarios and to determine one of the most important feedback loops in the

Arctic climate system. In addition, natural (including vegetation changes due to climate change) and human induced (management, urban development) land use changes will impact BVOC emissions. Within CRAICC Earth system modelling included detailed aerosol effects and basic physical aerosol processes as well as updated parameterizations for BVOC emissions. Model experiments were carried out to investigate the BVOC-aerosol-cloud-climate feedback loop. Idealized climate change simulations indicated significant increases in global monoterpene emissions,

emphasizing a potentially strong negative climate feedback mechanism especially over boreal forest. Nevertheless, future work needs to be done to better quantify BVOC emissions in a future climate. To build a deeper understanding of aerosol ageing processes, including long-range transport from Eurasia and North America to the Arctic, it will be necessary to obtain size segregated chemical composition information for sub-micrometre Arctic aerosol particles and VOCs. This information is also highly relevant for determining the CCN potential of aerosols.

A Lagrangian chemistry transport model was also implemented within CRAICC in order to evaluate how BVOC emissions from subarctic forest impact the formation of new particles and their growth into the CCN size range (Öström et al., 2017). New particles over the subarctic forest were found to be dominated by low-volatility HOMs formed from the ozonolysis and OH-oxidation of monoterpenes. Given the relatively low present day emissions of anthropogenic secondary aerosol precursors and primary particles in the subarctic forest region, new particle formation and subsequent

growth by HOMs most likely has an important role in maintaining CCN concentrations. However, because precursors involved in new particle formation are still not well described, the anthropogenic impact on new particle formation is not well constrained. This causes large uncertainties in the estimates for both preindustrial and future CCN concentrations in the Arctic and Subarctic. Thus, without more fundamental knowledge about the formation and initial growth of new particles and their role in maintaining CCN concentrations, the strength and importance of the BVOC-

aerosol-cloud-climate feedback loop remains poorly quantified.

The forecasted significant warming, consequent vegetation shifts, and ice-free Arctic waters all suggest that natural aerosol precursor emissions from the biosphere will increase substantially, although currently the magnitude of these effects is unknown (Bäck et al., 2012; Aalto et al., 2015; Schollert et al., 2016). In particular, if an increase in the height and cover of shrubs and graminoids occurs as a response to warming in the Arctic (Elmendorf et al 2012), there will



be a major impact on emission rates and consequently SOA formation (Tiiva et al. 2008; Faubert et al. 2010, Valolahti et al 2015, Kramshoj et al 2016). Conversely to temperature, increasing CO2 decreases BVOC (at least isoprenoid) emissions. However, using Earth system simulations, the combined impact of increasing (doubled) CO2 and temperature was shown to increase both BVOC emissions and aerosol number concentration by > 30% (Makkonen et

al., 2014). Furthermore, changes in aquatic ecosystems are known to affect biogenic emissions but the processes are not sufficiently understood to quantify (Faust et al., 2016).

Atmospheric pollutants are often reactive species that undergo continuous transformations in the gas-, particle-, and aqueous- phases implying that new compounds are formed and secondary aerosol mass is produced by gas-to-particle conversion or cloud processing. Anthropogenic pollution in the Arctic remains sourced primarily from long-range

transport, and its transformation through atmospheric chemistry and physics varies seasonally due to the position of the polar dome and due to the alternating absence and presence of solar radiation in the high Arctic. Thus, the endpoint of products also varies, with significant uncertainty due to the relatively long aging times and myriad exposure conditions. Furthermore, although many reaction schemes are well known, few are studied under extreme conditions at temperatures far below 0   C. In general, this leads to a lack of knowledge concerning the partitioning of species

between the gas and particle phase at Arctic and extreme conditions.

In most cases spatially resolved concentrations of atmospheric species are restricted to surface level measurements. In the Arctic information in the vertical dimension is largely missing due to the regions remoteness and harsh meteorological conditions. A number of modern techniques could promote Arctic data collection with vertical resolution, including UAVs, tethered balloons, and ground and satellite based remote sensing technologies. Aircraft

measurements also remain an option, but also expensive and only cover small time spans. Currently, such observations are not contributing to existing monitoring networks and are limited in time and space but the use of such technologies needs to be extended and promoted throughout the Arctic.

Although the state of scientific understanding regarding the direct forcing of greenhouse gases and atmospheric particles has developed, the level of understanding aerosol indirect forcing remains limited. In the high Arctic, even

aerosol direct forcing leaves open questions because the mixing state of long-range transported pollution is not sufficiently well described. The role of clouds and especially the impact of anthropogenically emitted particles on clouds is still uncertain although it has been the focus of intense research for decades. Indirect forcing of Arctic clouds is also unique because the polar night excludes cloud interactions with solar radiation, but does allow cloud interactions with terrestrial radiation. The Arctic haze season is also a unique anthropogenically influenced phenomenon and

requires deeper insight into its cloud impacts. More experimental work on the seasonal behavior of cloud radiation is needed and must be implemented into climate models. Work in CRAICC has been carried out to analyze the influence on the number of available CCN in a future climate based on the application of Earth system modeling. However, further quantification of indirect aerosol effects requires more fundamental understanding of the role of Arctic clouds and the anthropogenic influences on Arctic cloud formation.


A substantial part of Arctic warming has been attributed to BC deposition on snow-and ice-covered surfaces (Quinn et al., 2008). CRAICC has contributed to illuminating this scientific area of interest using measurements of atmospheric BC concentrations and deposition and comparing the observed values to modeled results (e.g. Massling et al., 2015; Ruppel et al., 2017). In addition, BC records retrieved from paleoclimate archives (lake sediments and ice cores) within

CRAICC have contributed to understanding the scale and significance of modern variations in BC with respect to historical values (Ruppel et al., 2014, 2015). However, estimates of how the spatial distribution of albedo will change due to future BC deposition remains highly uncertain due to limited knowledge that depends on many factors. For example, the growth of Arctic shipping and extraction of minerals and oil from Arctic reservoirs. Furthermore, the Arctic response to BC may be unique. For example, by comparing local BC emissions to mid-latitude emissions, Sand

et al. (2013a) found that local emissions have a much stronger surface temperature response than do mid-latitude emissions. In addition to the direct albedo response to BC, BC particles also affect snowpack albedo by changing the snow and ice crystal grain sizes, an effect that is not well described and is not included in climate models. The bottom line is that BC projections need to be improved for further model implementation.

Snow albedo varies spatially, temporally and spectrally, and is determined by snow properties (such as snow grain size

and shape, snow depth, age, cleanness) and the surrounding environment (e.g., weather and climate, topography, tree shades and penumbra). From these, the effective snow grain size, i.e., grain size and shape distributions, or specific surface area (Domine et al., 2006), is the most important parameter determining snow albedo. Surface darkening due to BC, dust, or other impurity deposition causes spectrally dependent albedo declines. Natural snow metamorphism processes constantly modify albedo, and when snow ages, with or without melting, snow grain sizes increase and as a

consequence albedo decreases (Wiscombe & Warren 1980). Future changes in temperature, rainfall, wind, humidity, cloudiness and/or depositing aerosols will therefore affect snow albedo. Extreme winds, in turn, can break snow grains into smaller entities and create snow dunes, or induce snow-dust storms (Dagsson-Waldhauserova et al. 2015), all of which significantly contributing to changes in snow albedo. Within CRAICC strong efforts were made to characterize the Icelandic dust reaching the high Arctic. The influence of dust on climate in the cryosphere has not been sufficiently

studied, although it may be on the same order of magnitude as the effect of BC on cryospheric surfaces. Experiments quantifying melting and insulation effects of dust layers on snow- and ice-covered surfaces were carried out within the CRAICC project (Dragosics et al., 2016; Wittmann et al., 2017). The frequency, variability and intensity of Icelandic dust events were investigated (Dagsson-Waldhauserova et al., 2013, 2014a) and the physical properties of Icelandic dust were determined (Dagsson-Waldhauserova et al., 2014b, 2015, 2016). Dusts' influence may also be amplified by

melting of Icelandic (Arctic) glaciers, which may enlarge the effect of dust resuspension.

With a rapid development of computing power, paleoclimate models are becoming increasingly useful tools to investigate past climate changes on various spatial scales. Climate models are mathematical representations of our understanding of the climate system, including movements of heat and mass within components of the climate system and also interactions between different components. Testing of different scenarios, termed sensitivity testing, allows us



to explore plausible mechanisms behind climate changes and to analyze temporal-spatial variability. All paleoclimate models include uncertainties, and the magnitude of the uncertainty in the model output depends not only on the forcings used in the simulations, but also on model-specific features, such as the physical principles, complexity and resolution. Multi-model comparisons and proxy-model comparisons provide the means to test the reliability of model performance.

The reliability of the models and simulations increase if independent models consistently indicate the same or similar results and if the model results agree with the proxy-based climate reconstructions. Such recent model tests have shown that the paleoclimate models generally indicate consistent results for the Holocene in northern Europe, including the Arctic, but differ substantially in other Arctic regions, such as eastern Siberia and Alaska (Zhang et al. 2017).

The influence of short-lived climate forcers in the high Arctic is still a highly uncertain quantity as it depends on natural
and anthropogenic emissions and their complex interactions. Both types of emissions may change in the future, but those changes may also be for different reasons. Anthropogenic emissions will change as a result of changed activity patterns and burdens at mid-latitudes and in industrialized areas - the regions where anthropogenic emissions observed in the high Arctic presently originate. Changes may involve e.g. introduction of new cleaner technology and changes of emission regulations. In general, better emissions projections are needed as they strongly determine the level of
pollution in the Arctic. Within CRAICC, a large amount of data from multiple research platforms was utilized in order to qualitatively and more importantly quantitatively asses the state of the knowledge of natural emissions and their changes.

Field scientists and climate modelers have worked closely together to advance the knowledge of many complex Arctic topics. The lack of extensive ground based monitoring in the Arctic promotes this kind of large international
collaboration between scientists that operate or use existing Arctic monitoring networks. In the Arctic, in addition to the direct effects of increasing global CO2, there is a large risk that an acceleration of the temperature increase will occur due to feedback processes that occur only in the Arctic. Continued Arctic research is needed to provide better parametrizations in global models, which can be used to identify risks and climate thresholds and thereby inform politics and policy making, and help to weigh climate adaptation versus climate mitigation.

**Acknowledgements**

CRAICC acknowledges the following institutions for financial support: the Finnish Cultural Foundation, Grant: Prof. Markku Kulmala "International Working Groups"; Russian Mega-Grant No. 11.G34.31.0048 (University of Nizhny Novgorod); Academy of Finland contracts 259537, 257411 and 254195; Beautiful Beijing (Finland- China collaboration project) funded by TEKES; Nordforsk CRAICC-PEEX (amendment to contract 26060); CRAICC-
CRUCIAL (project Nr. 81257); Icelandic Research Fund (Rannis) Grant No. 152248-051; Danish Environmental Protection Agency with means from the Dancea fund for environmental support to the Arctic Region (M 112 002700); the Villum Foundation; the Carlsberg Foundation (project 009_1_0515); COST1303 (TOPROF); COST ES1404 (HarmoSnow); Pan-Eurasian Experiment PEEX. The development and use of NorESM1 was supported by the



Norwegian Research Council through the projects EarthClim (207711/E10), EVA (grant no. 229771) , the NOTUR (nn2345k) and NorStore (ns2345k) projects, and through the Nordic Centre of Excellence eSTICC (57001) and the EU H2020 project CRESCENDO (Grant no. 641816). Further CRAICC wants to thanks Rogier Floors for providing Fig. 8 and Christoph Münkel for Fig. 9.

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





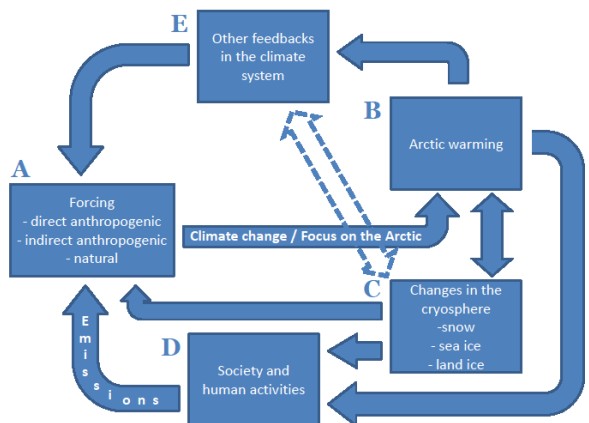

**Figure 1:** Interlinks and feedbacks in Arctic climate-cryosphere interaction. Capital letters indicate different components of the feedback systems and arrows emphasize the directions of interactions between components (unfilled arrow between component C and E indicates a previously undiscovered feedback loop in the system).

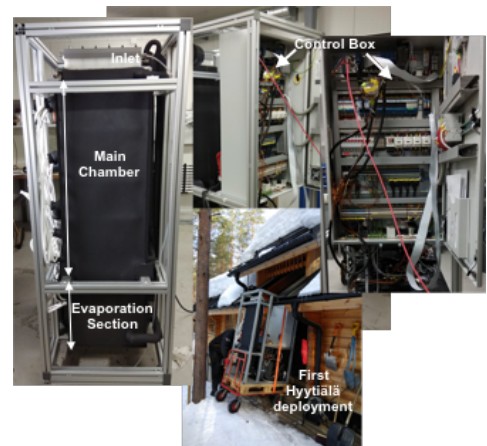

**Figure 2:** Images of the PINCii instrument with the main ice-coated flow reactor chamber, evaporation sections, and electronic control box indicated. The instrument was first deployed for field testing during the 2018 Hyytiälä HyICE measurement campaign.



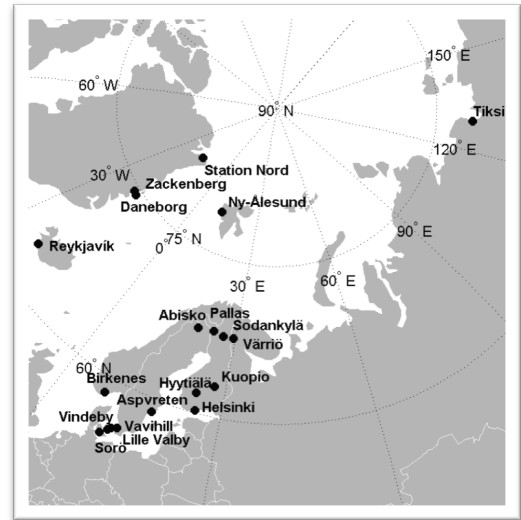

**Figure 3:** Map of the core field stations in CRAICC: Troll station, Antarctica, Vavihill, S-Sweden, Birkenes, S-Norway, Lille Valby, Denmark, Vindeby, Denmark, Sorø, Denmark, Aspvreten, central Sweden, SMEAR III, S-Finland, SMEAR II, central Finland, SMEAR IV, Kuopio, central Finland Sodankylä, N-Finland, SMEAR I, Värriö, N-Finland Abisko, N-Sweden, Pallas GAW station, N-Finland Tiksi, Siberia, Daneborg and Zackenberg, Greenland Ny-Ålesund, Spitzbergen, Villum Research Stagion, Greenland

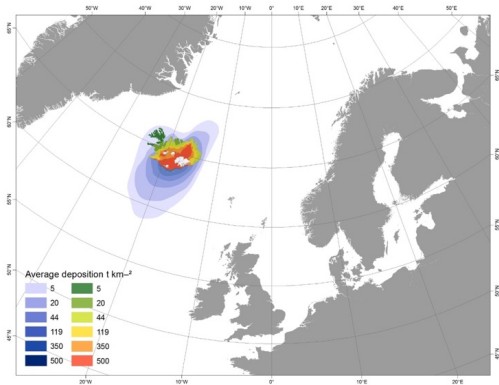

**Figure 4:** Annual dust deposition rates of Icelandic dust (Arnalds et al., 2014)



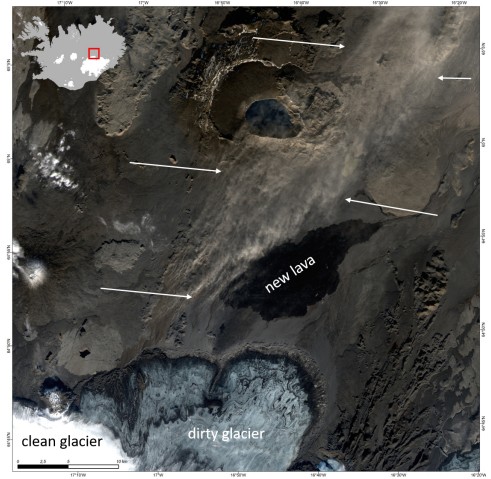

**Figure 5:** LANDSAT image of the Vatnajökull glacier (lower left corner, clean (dirty) glacier) polluted with dust (lower middle, dirty glacier) while a dust storm (brown dust plume indicated with white arrows) originating from the glacial flood plain Dyngjusandur over the new lava (dark surface in the center of the image) from the Holuhraun eruption 2014-2015. Source and date: NASA & USGS, 20th October 2016

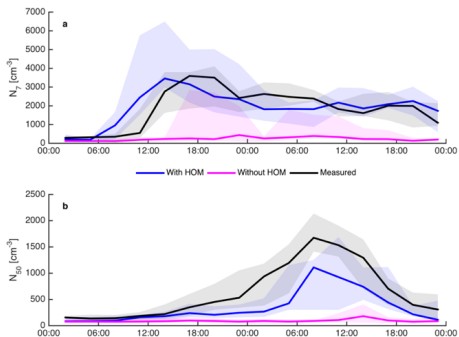

**Figure 6:** Modelled and measured (a) $N_7$ and (b) $N_{50}$. The model results are shown for simulations with and without HOM formation via autoxidation of monoterpenes (blue and pink lines respectively). The solid lines show the median values from 10 NPF events at Pallas. The shaded areas give the 25% to 75% interval.





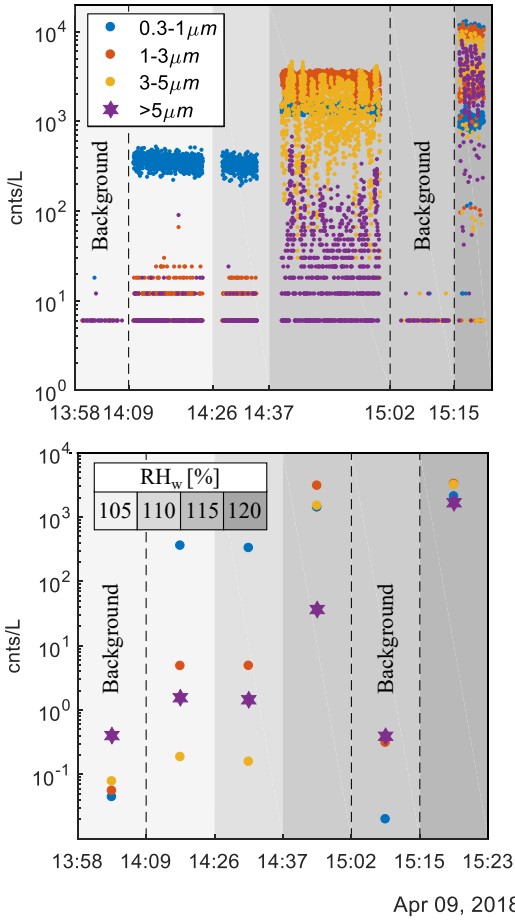

**Figure 7:** Initial field data of ice crystals per liter (cnts/L) collected using the PINCii chamber. A 4-channel OPC placed at the exit of the ice growth chamber is used to count particles exposed to various thermodynamic forcings illustrated using the chamber relative humidity with respect to water $RH_w$. The sample flow is initially directed through a 2.5 µm impactor to eliminate large particles and after the ice growth chamber the flow is exposed to an evaporation section where ice particles continue to grow while liquid droplets shrink. Thus, the largest size channel can be assumed to be made up solely of ice particles. Ice crystal formation is observed at $RH_w \geq 105$, while at the highest $RH_w \approx 120$ so-called "droplet breakthrough" is observed.




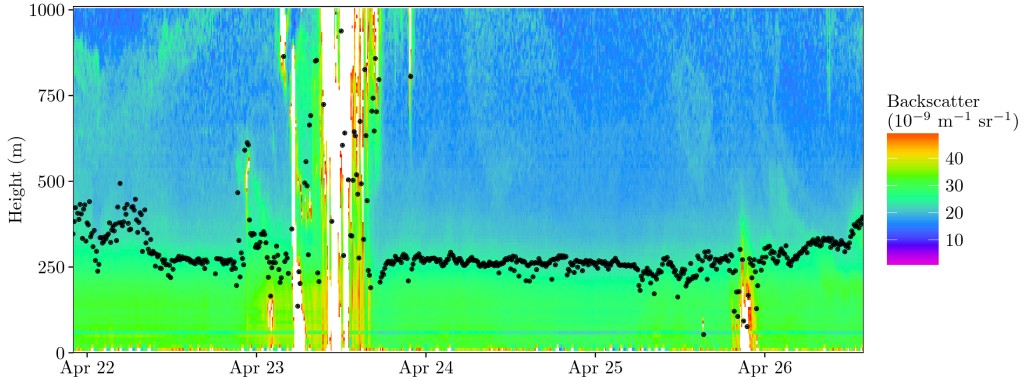

5 **Figure 8:** Backscatter profiles measured by a ceilometer from 22 to 27 April 2012 at the Villum Research Station, High Arctic. The bar to the right shows the backscatter density. The circles show the height of the boundary layer.

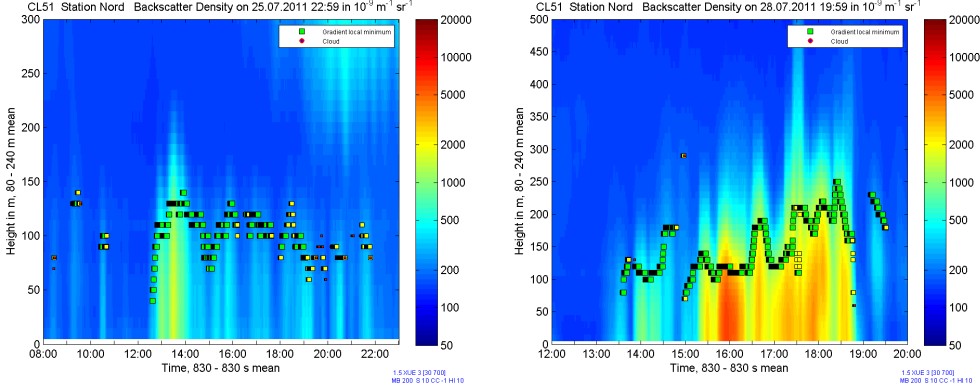

**Figure 9:** Backscatter profiles measured by a ceilometer on 25 July (left panel) and 28 July (right panel) in 2011. The bar to the right shows the backscatter density in $10^{-9}$ m$^{-1}$ sr$^{-1}$. The squares show the height of the boundary layer.



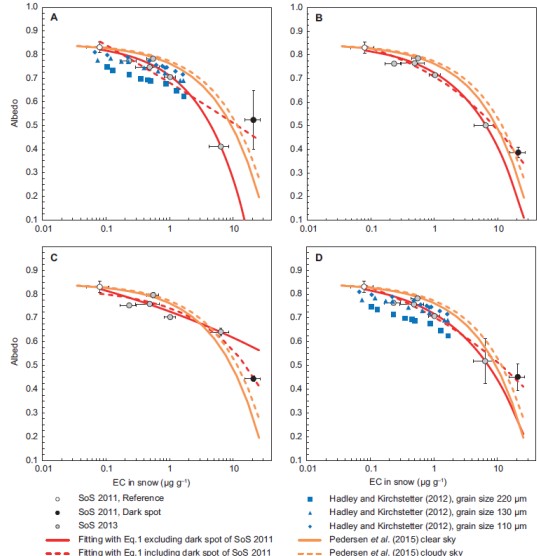

**Figure 10:** Broadband albedos at noon on days 1–3 as a function of the EC concentration in the surface layer: (A) day 1, (B) day 2, (B) day 3, and (D) days 1–3. In all plots, the SoS2011 reference albedo is the average over solar noon albedos during the first week of the experiment. In A–C, the gray and black circles are 60-minute albedo averages at solar noon, the vertical and horizontal bars are standard deviations for albedo and EC, respectively. In D, the circles are the albedo averages at solar noon of the days 1–3, and the vertical and horizontal bars are standard deviations for albedo and EC, respectively (Svensson et al., 2016).

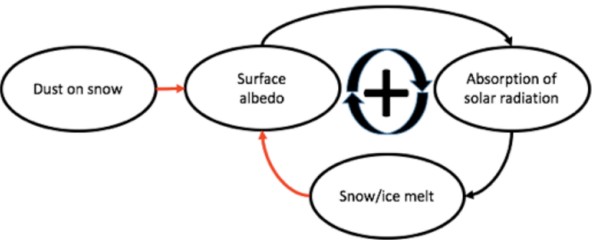

**Figure 11:** The feedback loop of the 'dust albedo effect' for Icelandic dust events (perturbation), a new hypothesis that Icelandic dust deposited on snow can be a cause for Arctic amplification via such positive feedback. Measurements show that black Icelandic dust particles are highly light absorbing similar to soot particles (Peltoniemi et al., 2015).



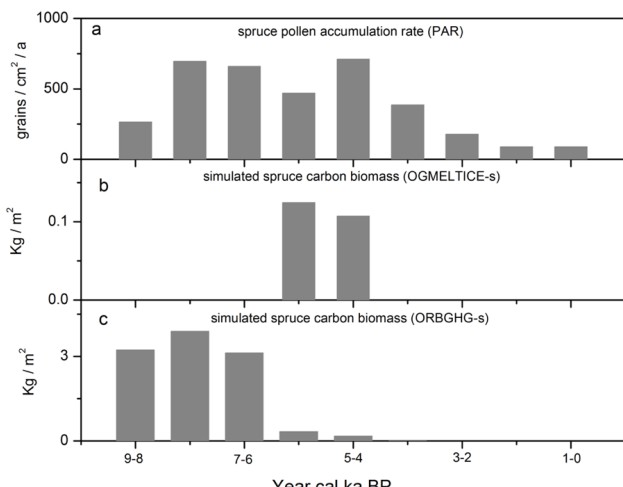

**Figure 12:** Time series comparisons between the pollen data from Kharinei Lake in the modern tundra (62.75°N,
5  67.37°E) and the simulated spruce biomass from the nearest gridcell (62.50°N, 67.00°E). a) Pollen accumulation rate
(PAR) data for spruce, b) The simulated spruce biomass based on paleoclimate scenario OGMELTICE-s, c) The
simulated spruce biomass based on paleoclimate scenario ORBGHG-s. The simulated biomass and PAR values are
shown as mean values of 1000-year intervals.

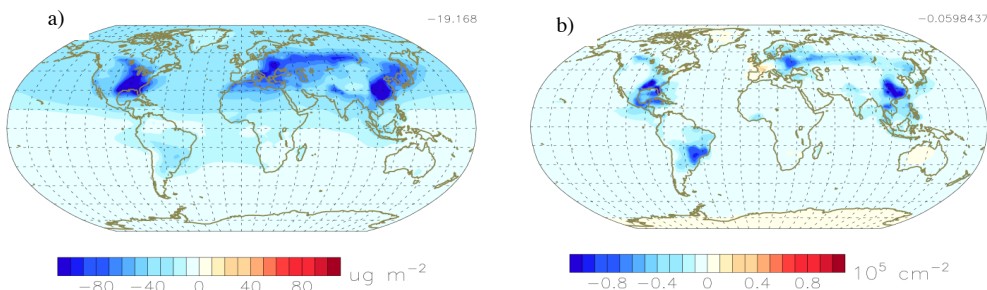

10  **Figure 13:** Simulated a) organic aerosol column burden change and b) cloud droplet concentration changes between
1850 and 2000; attributed to anthropogenic land use change (CMIP5).



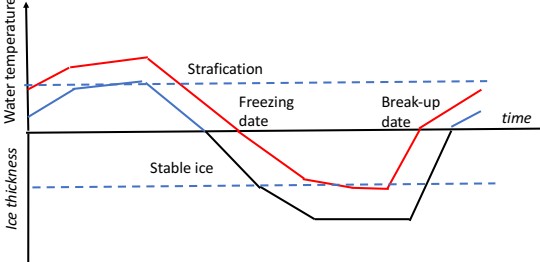

**Figure 14:** The annual cycle of boreal and tundra lake surface temperatures and ice thicknesses. The blue/black lines show a reference state with the red line illustrating climate warming impact. Ice and open water seasons change their length, including the presence of winter stable ice cover and/or summer stratification.

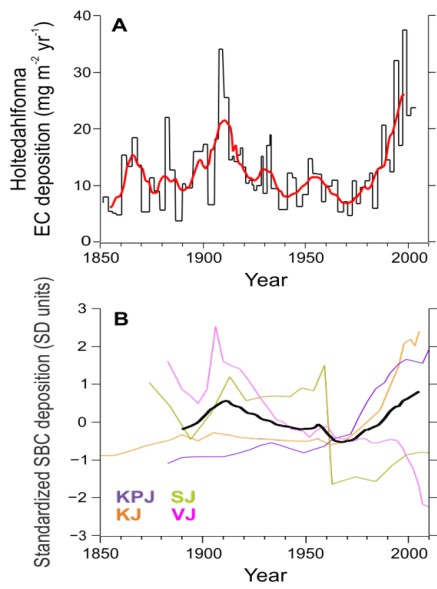

**Figure 15:** BC deposition at a Svalbard glacier and four northern Finland lakes between 1850 and 2010. A) EC
10   deposition at Holtedahlfonna and a 10-year running average (red line). B) Stacked SBC deposition at four northern Finland lakes (KPJ, KJ, SJ, and VJ) expressed as standard deviations from the mean. The black curve indicates a LOESS smoother (span of 0.15) of time intervals from which data is available from all the lakes. Adapted with permission from Ruppel et al. (2015). Copyright © 2015, American Chemical Society.





| Model simulation | Version | Land use | Oxidants | Boundary conditions (year) | Length |
|---|---|---|---|---|---|
| 1 | Mozart | 1850 | Online | Present-day Climatology (2000) | 12 years |
| 2 | Mozart | 2000 | Online | Present-day Climatology (2000) | 12 years |
| 3 | SOA | 1850 | Present-day Climatology (2000) | Present-day Climatology (2000) | 7 years |
| 4 | SOA | 2000 | Present-day Climatology (2000) | Present-day Climatology (2000) | 7 years |

**Table 1:** Simulations of aerosol-climate effects of anthropogenic land-use. Mozart refers to a version of NorESM1-CRAICC coupled with a tropospheric chemistry model.