# Peer review of "Interactions between the atmosphere, cryosphere and ecosystems at northern high latitudes"

_Atmospheric Chemistry and Physics, 2018_

## Referee Comment (RC1) · Anonymous Referee #1 · 3 Sep 2018

The paper addresses important issues, and is a summary of five or more years of research. It is a compendium on aerosol studies in general with special focus on high northern latitudes. Undoubtedly the paper will be of interest to many readers and will serve as a guide to future studies.

As fits my area of expertice I was going to focus my comments about this paper on the cloud active aerosol (CCN and INP) results. As it turns out these are two areas where the paper has little to say.

The CCN instrument is mentioned on page 13 line 15 but no results are reported. Potential CCN sources are discussed in sections about new particle formation and in modeling studies without supporting CCN measurements. Perhaps these will be forthcoming in later publications.

[Figure]

Instrument development for measurements of ice nucleating particles (INPs) is reported on pages 11 and 13, some results are described on page 24.-25. The new instruments represent advances toward INP studies, but the results are not yet in. The sample output from one of the devices in Fig. 7 is evidence that the PINCii instrument is functioning more or less as other similar devices do. NOt much more. Reagarding Fig.7, the upper and lower panels of Fig.7 are not identified, and seem to be the same data in two different formats. Temperature information is missing.

The same text is repeated iin lines 30-34 of page11 and lines 2-5 of page 24.

The laboratory studies described in 4.1.4.1 are important basic studies about water adsorption on surfaces. The link to the rest of the paper is rather abstract and the connection mentioned in lines 23-25 on page 24 is tenouos.

The increases in INP concentrations due to ship exhaust reported at the top of page 25 are interesting. However, it is not known whether the ship emissions would remain ice nucleation active on the longer time scale than the few minutes that was the case in the sampling used.

In all, the status of the CCN and INP data should be given in order to justify mention the instrument developments reported.

---

## Referee Comment (RC2) · Anonymous Referee #2 · 4 Nov 2018

General Comments.

The paper certainly gives a long missed overview about CRAICC and from that reason the paper is highly welcomed. The authors certainly are capable of writing the paper and I certainly do support the publication.

The authors list carefully all the topics treated so far in CRAICC. They point also to missing observations. They put emphasis to overlooked facts, like Iceland being a strong desert particle source. In the last century, such a paper would have been written by one or two authors with proper credit to all contributors. Today, that paper (45 pages text) has been written by some 63 authors, meaning ¾ pages of text for each of them!

However, that paper is missing an ordering hand. The specific comments and technical corrections will show that. Among those many authors, it should have been possible devoting one looking for formal mistakes. This would ease the work of the reviewers.

Specific Comments.

Page 4, line 14 and many other places. Air pollutants: this is a valuing term. Scientific papers should be without valuing terms (impurities, and others), except it is clearly proven, like "anthropogenic pollution". Atmospheric trace gases are no impurities, they are minor constituents.

Page 5, line 10 and many other places. The emissions of sea spray aerosol. Sea water contains not only water-soluble salts, it also contains large amounts of insoluble biological material with the potential of turning into aerosol particles if injected into the atmosphere. This biological material (microbes for instance) is mentioned in this paper only for snow, page 7, line 12 and on page 6, line 30 as "sea salt and primary organic aerosols".

*"A microorganism, or microbe, is a microscopic organism, which may exist in its single-celled form or in a colony of cells (Wikipedia). In the open ocean, far from the influences of coastal human habitation, sea water still contains huge numbers of microbes. Coastal areas can contain even greater concentrations. Vast numbers of bacteria and plankton occur both at the surface and in deep ocean waters. Viruses are entities that require bacteria or other cells in order to make copies of their genetic material and to construct new casings that house the genetic material. Scientific studies have shown that 10 to 100 million viruses can be present in a teaspoonful of sea water (Wikipedia).*

The concentration is about $5 \cdot 10^9$ microbes per liter or $5 \cdot 10^6$ cm$^{-3}$ (F. Azam, T. Fenchel, J. G. Field, J. S. Gray, L. A. Meyer-Reil and F. Thingstad (1983): The Ecological Role of Water-Column Microbes in the Sea. Marine Ecology). Blanchard, D.C.L.D. Syzdek (1972, Concentration of Bacteria in Jet Drops from Bursting Bubbles. J. Geophys. Res. 77, 5087-5099) have shown that the concentration of bacteria in rising and bursting bubbles might increase by a factor of 1000. Not to mention the contribution of Leck and Bigg to "biological" aerosol from Artic leads; and Cindy Morris (and the papers, she is pointing) to Biological Ice Nuclei at all. If all those primary biological particles are of minor contribution to the Arctic aerosol, that should be stated in this paper. Alternatively, it also should be stated, why primary biological particles are not treated in CRAICC!

Snow in the Arctic (below 0°C) behaves like dust in a desert. Parts of it are reinjected into the atmosphere by turbulent winds. If snow contains microbes (page 7, line 12), they are reinjected and are becoming part of the Arctic aerosol.

BTW: Bare lands (Iceland) are behaving the same way.

**You have capable biologists in your list of authors.**

Page 18, line 5. This critical statement about the performance of satellites should include that only atmospheric tracers with a rather long residence time (~ one week) should be observed, because of the orbit parameters and the swath width. That means, a full global coverage is only available after one week. In Polar Regions that might be different (page 21, line 8?).

Page 27, line 10. Description of Iceland as a desert is a duplication. See my remarks about an ordering hand.

Page 37, line 1. It should be indicated, how many days it's raining. If the precipitation is occurring only at a few days, dry deposition is more effective.

Page 38, line 22. "Legacy of CRAICC" has the taste of self-praise (self-praise is no recommendation). This number 5 could most probable be removed.

Page 40, line 19. What are "ageing time scales"?

Page 46, line 32. "Climate models are mathematical representations of our understanding of the climate system". I strongly disagree to that phrase. "Climate models are mathematical representations of our ability to model the climate system".

Technical Corrections.

Page 4, line 34. $CO_2$.

Page 5, line 4. IPCC 2013 is not in the reference list.

Page 7, line 31. 10-5 thousand years BP could be misunderstood. Better would be "five to ten thousand years BP".

Page 14, line 27. Instead of "formation" use better "production".

Page 15, line 30. The term "below cloud scavenging" is much more common.

Page 15, line 21. "The dispersion model considers BC as an inert pollutant with a size distribution described by a single size bin ranging from 0.001 to 1 µm in dry particle diameter" – This is no size distribution.

Page 18, line 14. 7x7 km2 is misleading and 13x24 km2 as well. Are you meaning 7x7 km? A better term would be "7 · 7 km".

Page 18, line 18. A number of commas are missing.

Page 19, line 29. "system perturbation". Better is "displacement", perturbation could have a valuing meaning.

Page 20, line 27. "1,000 µg m [next line] $^3$ during 24-hour intervals". The authors should look to a better handling of their word processors. This bad printing occurs, because the authors are not aware of the different hyphens. It should mean "1,000 µg m$^{-3}$"?

Page 23, line 7. Better "Secondary organic aerosol precursor sources".

Page 26, line 8. "important source of absorption" – look for a better term.

Page 27, line 3. What are "wet wind conditions"?

Page 29, line 18, 19: Better than 9 ka, 5 ka might be 9000, 5000 years BP.

Page 30, line 12. The paper should include errors. 0.09 to 0.10 mm h$^{-1}$ are very close together. That certainly is falling into the error bars.

Page 32, line 5. What is "organic sea spray"?

Page 35, line 30. A better term for "challenging" might be "questioning".

Page 41, line 25. "65% of the territory" – what territory? That of Russia?

Page 43, line 11. What is "uncovering of Arctic waters"?

Page 45, line 2, and 3. Better use $CO_2$.

Page 45, line 14. 0°C.

Page 45, line 18. Better than "meteorological" is "environmental". This page contains many judgmental terms.

Page 47, line 21. $CO_2$

The **5 References** should be kept uniform. Use something like EndNote.

Page 58, line 14. Omit line.

Page 63, line 11. Even if the journal recommends a citation, it should be kept in agreement with all the other references. On this page, many pages in the references are missing.

Page 65, line 26, Omit line break.

Page 66, line 34. Omit pp.

Page 67, line 13. Omit Vol.

Page 69, line 16. Pages of Atmos. Environ. are non-sense.

Page 69, line 24. Omit Vol., pp.

Page 71, line 34. Add authors.

Page 72, line 1. Left hand margin.

Page 75, Figure 4. The Figure should be blown up. The island Iceland would be sufficient. The units should be written as in the remainder of the paper.

Page 76, Figure 6. Much too small.

Page 77, Figure 7. Better than "cnts/L" might be "#/L". Why are the x-axis' equally spaced despite the fact, that the time steps are not.

Page 77, line 9. Give RH in %, $RH_w \geq 105\%$.

Page 78, Figure 9. Too small.
Page 79, Figure 10. Too small.
Page 80, Figure 12. Too small.
Page 80, Figure 13. Much too small

---

## Author Comment (AC1) · 13 Dec 2018

**Responds to the comments of the referees to the manuscript "Interactions between the atmosphere, cryosphere and ecosystems at northern high latitudes".**

We first want to thank both referees for their critical and constructive comments. We are aware that given the length of the manuscript it was a tremendous job to review this overview paper of the Nordic Centre of Excellence CRAICC. We have taken all of the comments and suggestions from the referees into consideration to improve the manuscript. Where our thoughts differ from those of the referees we provide statements as to why we believe the paper in its current form is correct or why we were not able to add the suggestions from the referees.

Following we first provide our comments to referee number one and then to number two. The text of the referees is provided in bold, our comments in normal black and in case we have made changes in the final version of the paper we have added this in red.

**Referee number 1**

**The paper addresses important issues, and is a summary of five or more years of research. It is a compendium on aerosol studies in general with special focus on high northern latitudes. Undoubtedly the paper will be of interest to many readers and will serve as a guide to future studies.**

**As fits my area of expertise I was going to focus my comments about this paper on the cloud active aerosol (CCN and INP) results. As it turns out these are two areas where the paper has little to say.**

**The CCN instrument is mentioned on page 13 line 15 but no results are reported. Potential CCN sources are discussed in sections about new particle formation and in modelling studies without supporting CCN measurements. Perhaps these will be forthcoming in later publications.**

CCN measurements are reported primarily within the context of the sea spray experiments. See for example sections 3.2.2 and 4.4.1.1. As we state in the "Summary and Outlook" (see p. 44, line 30 onwards) there remains a knowledge gap between BVOC emissions – new particle formation- and CCN. This area of active investigation clearly includes room for CCN measurements that remain to be undertaken.

**Instrument development for measurements of ice nucleating particles (INPs) is reported on pages 11 and 13, some results are described on page 24-25. The new instruments represent advances toward INP studies, but the results are not yet in. The sample output from one of the devices in Fig. 7 is evidence that the PINCii instrument is functioning more or less as other similar devices do. Not much more. Regarding Fig.7, the upper and lower panels of Fig.7 are not identified, and seem to be the same data in two different formats. Temperature information is missing.**

Figure 7 will be changed to better illustrate initial PINCii measurements. The caption will be modified accordingly, as well as some changes within the text when the figure is introduced:

[Figure]

"Figure 7: Initial data collected using the PINCii chamber. A 4-channel OPC placed at the exit of the ice growth chamber is used to count particles exposed to various thermodynamic forcing illustrated using the chamber relative humidity with respect to water $RH_w$; in all cases presented here the experimental temperature was -32° C. The sample flow is initially directed through a 2.5 μm impactor to eliminate large particles and after the ice growth chamber the flow is exposed to an evaporation section where ice particles continue to grow while liquid droplets shrink. In panel (a) an example of 15 min. incremented measurements of clean, background (BG) air and ambient air containing counted INPs ($D_p$ > 5 μm) at RH=105% is shown. The data points show the raw OPC data acquired as 5 s time averages of 1 s data, the black line depicts a 2 min. centered moving average, and the red lines are the 15 min. increment averages. The data is also illustrative of the slow increase of the chamber's background contribution over 2 to 3 hours, in line with existing CFDC systems. Panel (b) shows increment averages and is illustrative of the chamber's so-called "droplet breakthrough" when at the highest $RH_w \approx 120\%$ particle numbers in all size bins are amplified because liquid droplets no longer fully disappear within the evaporation section."

On Page 13 we will add to the instrument description starting at line 30:

"Manuscripts describing the PINCii instrument and the initial results of its first field deployment and ambient measurements are currently in preparation (see also Section 4.1.4; Castarède et al. in prep., Brasseur et al. 2018, Yusheng et al. 2018)."

To the text under section 4.1.4.2 we will add references to other CFDC instrument manuscripts to help to contextualize the PINCii development and again to refer to ongoing efforts to publish initial results from the initial PINCii field deployment:

"The INP counting and characterization instrumentation (Section 3.2) introduced to the Nordic region through the CRAICC collaboration was used in various field studies. The PEAC7 electrostatic deposition collectors and subsequent FRIDGE analysis (Schrod et al. 2016) were used to initiate the acquisition of building time series data for INP concentration in the Arctic environment, where very little observational data currently exists. A two-year time-series of measurements was established in Svalbard in conjunction with a global data series initiated through the EU FP-7 BACCHUS project (data currently being analysed). In an additional study the PEAC7 was used to sample sea-faring ship emission plumes transiting the Port of Gothenburg, Sweden. In those measurements, an amplification of ice nucleating capacity was observed from ship emissions (Thomson et al. 2018). Absolute INP activities observed from the ship plumes agreed with previous measurements of soot particles that suggest they are weak ice nucleators (Dymarska et al. 2006, Friedman et al. 2011, Schill et al. 2016). However, because these emissions are strongly concentrated, INP emission factors $10^5$–$10^7$ per kilogram fuel, they still result in strongly increasing INP compared to ambient values. As such the observed amplification may be important to consider as shipping routes open in Arctic waters and sea-ice loss increases (Peters et al. 2011, Fuglestvedt et al. 2014). The magnitude of cloud and climate scale effects will be a product of many additional factors, including how such INP observations may change with particle aging and how microphysical feedbacks and cloud response manifest in non-obvious ways (for example, see Possner et al. 2015).

In a very recent field campaign (HyICE2018) aimed at investigations of ice nucleation within the boreal environment the PINCii instrument developed within CRAICC was first field-deployed. In tests leading up to the campaign and during preliminary campaign data analysis the observed PINCii operational parameters and observed INP measurements are in good agreement with other instruments (Figure 7). In Figure 7 (a) the stability of the PINCii chamber background is illustrated, and in (b) initial tests of the conditions for water droplet breakthrough are shown to be in line with other existing CFDCs (cf., Stetzer et al. 2008, Garimella et al. 2016, Kong et al. 2018). During HyICE2018 data for instrument intercomparison was also collected using multiple existing INP measurement platforms (PINC, FRIDGE, SPIN, et al.). Campaign data analysis by all involved research groups is currently ongoing (cf., Castarède et al. in prep., Brasseur et al. 2018, Yusheng et al. 2018)."

**The same text is repeated in lines 30-34 of page11 and lines 2-5 of page 24.**

We thank the reviewer for noting the overlooked duplication. The page 24 duplication will be eliminated and the short introduction paragraph rewritten to better encompass the breadth of ice nucleation material covered in CRAICC:

"Field measurement campaigns have been initiated in an effort to quantify ice nucleating capacity of Arctic and Nordic air masses. Simultaneously idealized laboratory experiments have been

conducted to examine the underlying physicochemistry of molecular-scale processes from water adsorption to heterogeneous nucleation.  Results from such studies complement cloud-resolved modelling studies where CRAICC efforts have tried to connect the availability of ice nuclei to the production of ice crystals in clouds, advances which contribute to understanding how, for example, mixed-phase clouds form and evolve (Savre & Ekman 2015a, 2015b)."

**The laboratory studies described in 4.1.4.1 are important basic studies about water adsorption on surfaces. The link to the rest of the paper is rather abstract and the connection mentioned in lines 23-25 on page 24 is tenuous.**

The fundamental studies are illustrative of the range of detail and scales that were considered important in the context of CRAICC. The focus of some of these studies has been to seek detailed insight into molecular-level processes that contribute to macroscopic phenomena.  For example, the interplay between the kinetic and thermodynamic drivers of ice nucleation. The Arctic may be more sensitive to some of these things given its unique context. For example, there is a unique persistence of mixed-phase clouds in the Arctic that is unexpected from common reasoning (cf. Morrison et al. 2012).  The 4.1.4 sections (line 18 and onwards) have been rewritten to reflect these points with more clarity:

"Additional, laboratory investigations focused on how soluble salts may nucleate ice and whether or not at below eutectic temperatures sea salts might act as ice nucleating particles rather than deliquesce as CCN (Kong et al. 2018, Wagner et al. 2018, Castarède & Thomson 2018). All of these processes may affect cloud evolution and lifetime and thereby impact fundamental environmental processes like water cycle and radiative balance. Arctic clouds are known to be unique and often those that are most important to the regional radiative balance are mixed-phase clouds (Morrison et al. 2012), which include both liquid and solid cloud particles and therefore become inherently more unstable as temperatures diverge from the triple point (Savre & Ekman 2015a, 2015b).  Thus small effects, like those enumerated above, that influence the sensitive balance between liquid and solid cloud particle coexistence may have an amplified environmental feedbacks in Arctic air."

**The increases in INP concentrations due to ship exhaust reported at the top of page 25 are interesting. However, it is not known whether the ship emissions would remain ice nucleation active on the longer time scale than the few minutes that was the case in the sampling used.**

The referee makes a good point regarding time scale and the potential for changes with aerosol aging etc.  Changes to the text have been made (in line with the above discussion as well) and are included in the section 4.1.4.2 presented above, including multiple new references.

**In all, the status of the CCN and INP data should be given in order to justify mention the instrument developments reported.**

CCN measurements in CRAICC were made using established methods and thus the instrumentation is only briefly described within the text. Those studies specific to CCN are referenced.

Development of INP instrumentation undertaken within CRAICC has been a more laborious task and even now further development continues. Multiple manuscripts including a PINCii instrument manuscript are currently in preparation.  Another manuscript will cover an initial PINCii field

deployment and intercomparison with other ice nucleating particle measurement systems. Explicit statements in this regard will be added to the manuscript. See also our earlier responses.

*Referee number 2*

**General Comments**
**The paper certainly gives a long missed overview about CRAICC and from that reason the paper is highly welcomed. The authors certainly are capable of writing the paper and I certainly do support the publication. The authors list carefully all the topics treated so far in CRAICC. They point also to missing observations. They put emphasis to overlooked facts, like Iceland being a strong desert particle source. In the last century, such a paper would have been written by one or two authors with proper credit to all contributors. Today, that paper (45 pages text) has been written by some 63 authors, meaning 3⁄4 pages of text for each of them!**

**However, that paper is missing an ordering hand. The specific comments and technical corrections will show that. Among those many authors, it should have been possible devoting one looking for formal mistakes. This would ease the work of the reviewers.**

**Specific Comments**
**Page 4, line 14 and many other places. Air pollutants: this is a valuing term. Scientific papers should be without valuing terms (impurities, and others), except it is clearly proven, like "anthropogenic pollution". Atmospheric trace gases are no impurities, they are minor constituents.**

We partly agree with the referee that 'Air pollutants' is a valuing term, however, it is widely applied in the scientific community (5 titles in our publication list include the term) and seems to be accepted by many editors and journals. The sentence on page 4, line 14 "Importantly, these SLCFs are also air pollutants and in general their climate and air quality impacts must be simultaneously assessed" refers to a manuscript by Stohl and co-workers (2015) and there it is written on line 3 of the abstract: "ECLIPSE had a unique systematic concept for designing a realistic and effective mitigation scenario for short-lived climate pollutants." As our paper presents an overview of CRAICC activities and in this way also summarises the outcomes of many publications initiated through CRAICC we have applied certain phrases to be in line with the terminology of published papers.

**Page 5, line 10 and many other places. The emissions of sea spray aerosol. Sea water contains not only water-soluble salts, it also contains large amounts of insoluble biological material with the potential of turning into aerosol particles if injected into the atmosphere. This biological material (microbes for instance) is mentioned in this paper only for snow, page 7, line 12 and on page 6, line 30 as "sea salt and primary organic aerosols". "A microorganism, or microbe, is a microscopic organism, which may exist in its single-celled form or in a colony of cells (Wikipedia). In the open ocean, far from the influences of coastal human habitation, sea water still contains huge numbers of microbes. Coastal areas can contain even greater concentrations. Vast numbers of bacteria and plankton occur both at the surface and in deep ocean waters. Viruses are entities that require bacteria or other cells in order to make copies of their genetic material and to construct new casings that house the genetic material. Scientific studies have**

**shown that 10 to 100 million viruses can be present in a teaspoonful of sea water (Wikipedia). The concentration is about 5 $10^9$ microbes per liter or 5 $10^6$ cm$^{-3}$ (F. Azam, T. Fenchel, J. G. Field, J. S. Gray, L. A. Meyer-Reil and F. Thingstad (1983): The Ecological Role of Water-Column Microbes in the Sea. Marine Ecology). Blanchard, D.C.L.D. Syzdek (1972, Concentration of Bacteria in Jet Drops from Bursting Bubbles. J. Geophys. Res. 77, 5087-5099) have shown that the concentration of bacteria in rising and bursting bubbles might increase by a factor of 1000. Not to mention the contribution of Leck and Bigg to "biological" aerosol from Artic leads; and Cindy Morris (and the papers, she is pointing) to Biological Ice Nuclei at all. If all those primary biological particles are of minor contribution to the Arctic aerosol, that should be stated in this paper. Alternatively, it also should be stated, why primary biological particles are not treated in CRAICC!**

CRAICC initiated the construction of a temperature-controlled sea spray aerosol simulation tank to study sea spray's impact on the atmospheric aerosol budget, including the influence of organic material. The first research using the tank was directed towards a new sea salt source parameterization (Salter et al., 2014 and 2015). However, as these experiments utilized only artificial salt water, we cannot say anything about organic material. Lately we have performed experiments in the Stockholm tank including bacteria (funded within a new project), but that remains to be published.

In general, budget limitations forced us to limit CRAICC work mostly to sea salt sea spray. The strong influence of water temperature on sea spray emissions found by Salter et al (2014, 2015) suggests this to be an important, perhaps the most important, climate feedback path of sea spray. Climate effects of wind changes or of organic sea spray are much more speculative and need to be studied further.

Concerning the ESM activities in CRAICC we can say that primary biological particles are indeed treated in NorESM1, although in a simplified manner. These biogenic OM emissions have as a first approximation been given the same spatial distribution as the fine mode sea salt emissions (Kirkevåg et al., 2013, hereafter K13), with a global total of 8 Tg yr$^{-1}$, based on Spracklen et al. (2008). For comparison, the fossil fuel OM emissions for the year 2006 simulation in K13 amount to 6.3 Tg yr$^{-1}$, so 8 Tg yr$^{-1}$ is not a small contribution at all. To illustrate this, one of the test simulations in K13 (named natOMocn, see table 5 in K13) run without any biogenic OM from oceans (including a contribution from MSA) gave a 34% smaller total OM burden and a 13% smaller cloud droplet number concentration (CDNC) at 870 hPa in the pre-industrial simulation, globally and annually averaged. Because the indirect effect is sensitive to the background aerosol (through CDNC), the omission of biogenic oceanic OM (in natOMocn) was shown to yield an increase in the indirect radiative forcing between pre-industrial and present-day by as much as 38% (-0.46 W m$^{-2}$).

**Snow in the Arctic (below 0°C) behaves like dust in a desert. Parts of it are reinjected into the atmosphere by turbulent winds. If snow contains microbes (page 7, line 12), they are reinjected and are becoming part of the Arctic aerosol.**
**BTW: Bare lands (Iceland) are behaving the same way.**
**You have capable biologists in your list of authors.**

We do not agree with the referee on this point.  Snow (not only in the Arctic) is not like desert dust, although both are mineral systems.  Desert dust is generally made up of mineral material that exists very far from its melting temperature.  By comparison snow always exists very close to

its melting temperature, and thus is an active matrix material.  Its surfaces become more and more disordered as it approaches melting, leading it to become `sticky' due to the presence of liquid or liquid-like water.  The presence of this liquidity also allows for impurity transport processes on timescales that are significantly faster than what would be expected for a solid material (Bartels-Rausch et. al., 2012; 2014).

**Page 18, line 5. This critical statement about the performance of satellites should include that only atmospheric tracers with a rather long residence time (~ one week) should be observed, because of the orbit parameters and the swath width. That means, a full global coverage is only available after one week. In Polar Regions that might be different (page 21, line 8?).**

We do not agree with the reviewer on this point. On p18 we clearly state on line 8 that we are talking about atmospheric composition, and give examples. For current sensors this means that we have near global coverage each day (e.g. TROPOMI, MODIS, VIIRS, GOME-2), whereas for greenhouse gas sensors such as OCO-2 the swath is a limiting factor and the re-visiting time is 16 days. However, patterns in the spatial distribution and concentrations of $CO_2$ can be well-observed and the data provide information on local sources and $CO_2$ consumption during the growth season.

This is in contrast to the sensors mentioned on p21. Which provide high resolution imagery of land surface properties and relates to very different applications than atmospheric composition measurements.

We will extend the text on p. 18 to include the above in GHG observations and mention the near-global coverage. Thus, after the sentence: "For atmospheric composition, different instrument platforms are used to detect trace gases, greenhouse gases, and aerosols and clouds." we insert:

 "These include sensors such as TROPOMI, MODIS, VIIRS and GOME-2 with daily global coverage with spatial resolutions varying from 250 m for aerosols and clouds (MODIS and VIIRS) to 3.5 km x 7 km for methane, $NO_2$, $SO_2$, etc. from TROPOMI. Other sensors such as SLSTR provide smaller but still useful coverage especially at high latitudes. Together with information on forest fires and on aerosol absorption such data provide information on emission and transport of aerosols and trace gases and their effects on air quality, climate and pathways to the Arctic. $CO_2$ and SIF (Solar Induced chlorophyll Fluorescence) are retrieved from OCO-2 data with a revisit time of 16 days because of the rather narrow swath (10 km); the OCO-2 pixel size is 1.3 km x 2.2 km. Thus the spatial resolution of OCO-2 allows an anthropogenic $CO_2$ signature to be distinguished (Hakkarainen et al., 2016; Eldering et al., 2017). As for other GHG satellites, Tansat is comparable to OCO-2 in resolution, and GOSAT has a revisit time of only 3 days but a pixel diameter of 10 km."

**Page 27, line 10. Description of Iceland as a desert is a duplication. See my remarks about an ordering hand.**

We will delete the sentence:

"Iceland is the largest and most active high-latitude dust source, where dust deposition is expected to influence an area larger than 500,000 km2 (Arnalds et al., 2014; Dagsson-Waldhauserova et al., 2014a)."

**Page 37, line 1. It should be indicated, how many days it's raining. If the precipitation is occurring only at a few days, dry deposition is more effective.**

There are no AWS (automatic weather stations) at the core location but based on other field information we know that there is year round precipitation and therefore we assume that dry deposition is not a dominate process. Most precipitation will come as snow and the number of rain days are difficult to determine. Based on an AWS located approximately 400 m lower in elevation on the same glacier, there are temperatures above freezing during June-October so there is of course the possibility of rain at the coring location.

The model meteorology is based on ERA that is heavily assimilated by measurements. When we compared the model output with field observations the comparison showed reasonable agreement considering the resolution of the model.

**Page 38, line 22. "Legacy of CRAICC" has the taste of self-praise (self-praise is no recommendation). This number 5 could most probable be removed.**

It is unfortunate that the referee's impression of Section 5 was primarily that of self-praise. Although we certainly want to communicate the successes of CRAICC, it was not our primary intent to be self-edifying. Indeed our main aim for this section was to provide an overview of the strategy of CRAICC with regards to achieving a holistic understanding of the various feedback loops investigated. Beyond the scientific successes CRAICC was a successful execution of a multi-scale, multi-national research effort. We believe that such information could be useful for others who aim to study similarly complex systems and the two selected examples highlight how connections were made. This may not be so obvious by simply trying to follow the publication trail.

**Page 40, line 19. What are "ageing time scales"?**

"Ageing time scales" is a commonly applied phrase to define time periods over which aerosols are chemically and physically processed or changed while residing in the atmosphere. One example of such a process is the change of hygroscopicity vis-á-vis particle phase chemical reactions.

**Page 46, line 32. "Climate models are mathematical representations of our understanding of the climate system". I strongly disagree to that phrase. "Climate models are mathematical representations of our ability to model the climate system".**

We agree with the referee and will change the text on page 46, line 32 accordingly:

"Climate models are mathematical representations of our ability to model the climate system"

**Technical Corrections**

**Page 4, line 34. $CO_2$.**

Page 4, line 34 will be changed to:

"$CO_2$"

**Page 5, line 4. IPCC 2013 is not in the reference list.**

We will add IPCC 2013 to the reference list as:

"IPCC, 2013: Climate Change 2013: The Physical Science Basis. Contribution of Working Group I to the Fifth Assessment Report of the Intergovernmental Panel on Climate Change [Stocker, T.F., D. Qin, G.-K. Plattner, M. Tignor, S.K. Allen, J. Boschung, A. Nauels, Y. Xia, V. Bex and P.M. Midgley (eds.)]. Cambridge University Press, Cambridge, United Kingdom and New York, NY, USA, 1535 pp, doi:10.1017/CBO9781107415324."

**Page 7, line 31. 10-5 thousand years BP could be misunderstood. Better would be "five to ten thousand years BP".**

We will change page 7, line 30 and 31 to:

"One such useful climatic period is the Holocene thermal maximum about five to ten thousand years BP,"

**Page 14, line 27. Instead of "formation" use better "production".**

We do not agree with the referee here. The title of the publication from Hermansson et al. 2014, which is the reference for the model is: 'Biogenic SOA formation through gas-phase oxidation and gas-to-particle partitioning - a comparison between process models of varying complexity.'

**Page 15, line 30. The term "below cloud scavenging" is much more common.**

We assume that the referee is pointing to page 15, line 20 and will change the sentence:

"Wet deposition distinguishes between below- and in-cloud scavenging by both rain and snow"

**Page 15, line 21. "The dispersion model considers BC as an inert pollutant with a size distribution described by a single size bin ranging from 0.001 to 1 µm in dry particle diameter" – This is no size distribution.**

We agree with the referee and will been rephrase the sentence to clarify the way aerosols simulated in the model. We will exchange the sentence:

"The dispersion model considers BC as an inert pollutant, with a size distribution described by a single size bin ranging from 0.001 to 1 µm in dry particle diameter. Total particle production is integrated over each size bin (non-BC aerosol utilize full bin distributions) while dry and wet removal rates are calculated using mass-weighted mean diameter in each size bin."

with

"BC and other fine anthropogenic PM components are modelled as inert aerosol of 0.5 µm dry size. Emissions from natural sources (fires, sea-salt and desert dust) are parameterized in terms of continuous distributions and split into species-specific size bins. Deposition and settling of each bin are related to the mass-mean wet diameter of the bin."

**Page 18, line 14. 7x7 km2 is misleading and 13x24 km2 as well. Are you meaning 7x7 km? A better term would be "7 · 7 km".**

We will change 7x7 km2 and 13x24 km2 to:

"7 km x 7 km" and "13 km x 24 km"

**Page 18, line 18. A number of commas are missing.**

We will change the text on page 18, line 18 to:

"Information on the trace gases, such as $NO_2$, $SO_2$, formaldehyde (a proxy for less volatile organic compounds) and near-surface UV radiation is also important for the formation of aerosol particles, through gas-to-particle conversion."

**Page 19, line 29. "system perturbation". Better is "displacement", perturbation could have a valuing meaning.**

We will change the sentence on page 19, line 29 to:

"In general, quantifying a feedback loop requires the observation of the dampening (negative) or strengthening (positive) of a system displacement."

**Page 20, line 27. "1,000 µg m [next line] $^3$ during 24-hour intervals". The authors should look to a better handling of their word processors. This bad printing occurs, because the authors are not aware of the different hyphens. It should mean "1,000 µg m$^{-3}$"?**

We will ensure that in the final version the hyphens are typeset correctly and change it on page 20, line 27 to:

"1,000 µg m$^{-3}$"

**Page 23, line 7. Better "Secondary organic aerosol precursor sources".**

We agree and will change on page 23, line 7 to:

"Secondary organic aerosol precursor sources and processes …"

**Page 26, line 8. "important source of absorption" – look for a better term.**

Connected to this the authors realized that the acronym OC was not defined anywhere. It was mentioned for the first time on page 7 line 12 in the sentence

"Aerosol particles and other impurities in snow, including BC, OC, dust and microbes, also affect snow albedo and melt."

BC was defined already earlier (page 4, line 24) so we add the words organic carbon and the sentence on page 7 line 12 should read:

"Aerosol particles and other impurities in snow, including BC, organic carbon (OC), dust and microbes, also affect snow albedo and melt."

The sentence on page 26 line 8 now reads

"The results revealed that the OC in the snow could be an important source of absorption. "

The reviewer wishes that the words " important source of absorption " be changed. We agree and the sentence will be changed to:

" The results revealed that the OC in the snow could have a significant contribution to light absorption. "

By reading this section again we located yet another error. The sentence on page 26 lines 9 – 10 read:

"The OC absorption may also partly explain the high mass absorption cross section (MAC) value needed for BC to match the measured low albedo values with the SNICAR (Flanner et al. 2007) simulated albedo. "

The error is that there is no "OC absorption". Instead, there is "light absorption by OC". The corresponding sentence should be changed to

"Light absorption by OC may also partly explain the high mass absorption cross section (MAC) value needed for BC to match the measured low albedo values with the SNICAR (Flanner et al. 2007) simulated albedo."

**Page 27, line 3. What are "wet wind conditions"?**

The sentence page 27, line3 will be changed to 'moist and low wind conditions' as in the title of the referenced article. We refer to our dust storm monitoring when dust storms were captured during high precipitation season or when the main driver for dust suspension was not wind but solar radiation and surface heating.

"Dust aerosols interact with rain, snow and ice during dust storms. Suspended dust was observed during moist and low wind conditions"

**Page 29, line 18, 19: Better than 9 ka, 5 ka might be 9000, 5000 years BP.**

We will change on page 29, line 17 to:

"In Figure 12 northward expansion towards the current tundra is indicated by the rise of spruce pollen values 9000 years BP, and the late-Holocene retreat by the decline of spruce pollen values after 5000 years BP."

**Page 30, line 12. The paper should include errors. 0.09 to 0.10 mm h$^{-1}$ are very close together. That certainly is falling into the error bars.**

On page 30, line 12 we will include the standard deviations to put the small-looking changes into context:

"Indeed, cloud cover over boreal forest increases from 54.1 % to 56.9 % (0.5% standard deviation) and mean precipitation intensity increases slightly from 0.09 to 0.10 mm h$^{-1}$ (0.0017 standard deviation)."

**Page 32, line 5. What is "organic sea spray"?**

Organic sea spray is the organic fraction of sea spray due to organic surfactants on the sea surface and bubble surfaces, which follows the salt and water to the droplet surface. It's a well-established

concept (see for example many papers by Christina Facchini). It is not the same as biological sea spray. We are well aware that for example bacteria and viruses also stick to these surfaces and follow the sea spray.

**Page 35, line 30. A better term for "challenging" might be "questioning".**

We agree with the referee and will change on page 35, line 30 the second part of the sentence to:

"…, questioning the prevailing conception of declining or stable Arctic BC values during the last decades."

**Page 41, line 25. "65% of the territory" – what territory? That of Russia?**

The sentence will be changed to:

"The understanding and controlling of GHG and SLCF emission(s) and concentration dynamics over the Russian Arctic-boreal regions and the stability of Russian permafrost, currently occupying 65% of the Russian national territory (Melnikov et al. 2018), is one of the greatest scientific and social challenges for the entire northern Eurasian region."

**Page 43, line 11. What is "uncovering of Arctic waters"?**

The sentence:

The uncovering of Arctic waters was a major area of emphasis within CRAICC.

will be rephrased to:

"The loss of Arctic sea-ice was a major area of emphasis within CRAICC."

**Page 45, line 2, and 3. Better use $CO_2$.**

Will be changed to:

"$CO_2$"

**Page 45, line 14. 0°C.**

Will be changed to:

"0°C"

**Page 45, line 18. Better than "meteorological" is "environmental". This page contains many judgmental terms.**

The sentence will be changed to:

"In the Arctic, information in the vertical dimension is largely missing due to the regions remoteness and harsh environmental conditions."

**Page 47, line 21. $CO_2$**

Will be changed to:

"$CO_2$"

**The 5 References should be kept uniform. Use something like EndNote.**

**Page 58, line 14. Omit line.**

The extra line will be deleted!

**Page 63, line 11. Even if the journal recommends a citation, it should be kept in agreement with all the other references. On this page, many pages in the references are missing.**

We will change the reference to:

"Mårtensson, E. M., Nilsson, E. D., de Leeuw, G., Cohen, L. H. and Hansson, H.-C.: Laboratory simulations and parameterization of the primary marine aerosol production, J. Geophys. Res., 108, D9, 4297-4308, DOI: 10.1029/2002JD002263, 2003."

Also, we will add pages to the following references:

"Melnikov, V., Gennadinik, V., Kulmala, M., Lappalainen, H. K., Petäjä, T., and Zilitinkevich, S.: Cryosphere: a kingdom of anomalies and diversity, Atmos. Chem. Phys., 18, 6535-6542, 2018."

"Ng, N. L., Herndon, S. C., Trimborn, A., Canagaratna, M. R., Croteau, P. L., Onasch, T. B., Sueper, D., Worsnop, D. R., Zhang, Q., Sun, Y. L. and Jayne, J. T.: An Aerosol Chemical Speciation Monitor (ACSM) for Routine Monitoring of the Composition and Mass Concentrations of Ambient Aerosol, Aerosol Science and Technology Vol. 45 (7), 780-794, 2011."

"Nguyen, Q. T., Kristensen, T. B., Hansen, A. M. K., Skov, H., Bossi, R., Massling, A., Sørensen, L. L., Bilde, M., Glasius, M., and Nøjgaard, J. K.: Characterization of humic-like substances in Arctic aerosols, Journal of Geophysical Research: Atmospheres, 119, 5011-5029. 10.1002/2013JD020144, 2014b."

**Page 65, line 26, Omit line break.**

The extra line will be deleted!

**Page 66, line 34. Omit pp.**

We will delete:

"pp"

**Page 67, line 13. Omit Vol.**

We will delete:

"Vol."

**Page 69, line 16. Pages of Atmos. Environ. are non-sense.**

Will change the reference to:

"Sofiev, M., Siljamo, P., Valkama, I., Ilvonen, M., and Kukkonen, J.: A dispersion modelling system SILAM and its evaluation against ETEX data, Atmos. Environ., 40, 674–685, 2006."

**Page 69, line 24. Omit Vol., pp.**

We will delete:

"Vol."

**Page 71, line 34. Add authors.**

The authors have been added:

"Wanninkhof, R., Park, G.-H., Takahashi, T., Sweeney, C., Feely, R., Nojiri, Y., Gruber, N., Doney, S. C., G., McKinley, A., Lenton, A., Le Quéré, C., Heinze, C., Schwinger, J., Graven, H., and Khatiwala, S.: Global ocean carbon uptake: magnitude, variability and trends, Biogeosciences, 10, 1983-2000, 2013."

**Page 72, line 1. Left hand margin.**

We will change to the correct format!

**Page 75, Figure 4. The Figure should be blown up. The island Iceland would be sufficient. The units should be written as in the remainder of the paper.**

The figure has been improved accordingly:

[Figure]

**Page 77, Figure 7. Better than "cnts/L" might be "#/L". Why are the x-axis' equally spaced despite the fact, that the time steps are not.**

See the new Figure 7, and discussion in response to Referee #1.

**Page 77, line 9. Give RH in %, $RH \geq 105\%$.**

See the revised Figure 7 caption presented in the response to Referee #1.

**Page 76, Figure 6. Much too small. Page 78, Figure 9. Too small. Page 79, Figure 10. Too small. Page 80, Figure 12. Too small. Page 80, Figure 13. Much too small**

The size of the figures will change when they will be printed in the final version. All the figures are available for submitting to the journal with high resolution but were included in the ACPD-file only with low resolution to make it easier for the referees to download the file.

**References**

Brasseur, Z., Wu, Y., Castarède, D., Ahonen, L., Thomson, E.S., Kulmala, M., Petäjä, T. and Duplissy, J.: Development and operation of the ice nucleation chamber PINCII during the ice nucleation campaign held in Hyytiälä, Finland. The Centre of Excellence in Atmospheric Science - From Molecular and Biological processes to The Global Climate Annual Seminar, Kuopio, Finland Nov. 27-29, 2018.

Castarède, D., Brasseur, Z., Wu, Y., Duplissy, J., Thomson, E.T.: Updating the Zurich Portable Ice Nucleation Chamber, *manuscript in preparation.*

Dymarska, M., Murray, B. J., Sun, L., Eastwood, M. L., Knopf, D. A., and Bertram, A. K.: Deposition ice nucleation on soot at temperatures relevant for the lower troposphere, Journal of Geophysical Research – Atmospheres, 111, D04204, doi: 10.1029/2005JD006627, 2006.

Eldering, A., Wennberg, P. O., Crisp, D., Schimel, D. S., Gunson, M. R., Chatterjee, A., Liu, J., Schwandner, F. M., Sun, Y., O'Dell, C. W., Frankenberg, D., Taylor, T., Fisher, B., Osterman, G. B., Wunch, D., Hakkarainen, J., Tamminen, J., and Weir, B.: The Orbiting Carbon Observatory-2 early science investigations of regional carbon dioxide fluxes, Science, 358, doi:10.1126/science.aam5745, 2017.

Friedman, B. et al. Ice nucleation and droplet formation by bare and coated soot particles. Journal of Geophysical Research-Atmospheres 116, D17203, doi: 10.1029/2011JD015999, 2011.

Hakkarainen, J., Ialongom I., and Tamminen, J.: Direct space-based observations of anthropogenic $CO_2$ emission areas from OCO-2, Geophys. Res. Lett., 43, 11400-11406, doi:10.1002/2016GL070885, 2016

Morrison, H., de Boer, G., Feingold, G., Harrington, J., Shupe, M. D., and Sulia, K.: Resilience of persistent arctic mixed-phase clouds, Nature Geosci, 5(1), 11–17, doi: 10.1038/ngeo1332, 2012.

Savre, J., and Ekman, A. M. L.: A theory-based parameterization for heterogeneous ice nucleation and implications for the simulation of ice processes in atmospheric models, J. Geophys. Res. Atmos., 120, 4937 – 4961, doi:10.1002/2014JD023000, 2015a.

Savre, J., and Ekman, A. M. L.: Large-eddy simulation of three mixed-phase cloud events during ISDAC: Conditions for persistent heterogeneous ice formation, J. Geophys. Res. Atmos., 120, 7699–7725, doi:10.1002/2014JD023006, 2015b.

Schill, G. P. et al. Ice-nucleating particle emissions from photochemically aged diesel and biodiesel exhaust. Geophysical Research Letters, 5524–5531. doi: 10.1002/2016GL069529. 2016.

Stetzer, O., Baschek, B., Lueoeond, F., and Lohmann, U.: The zurich ice nucleation chamber (zinc) - a new instrument to investigate atmospheric ice formation, Aerosol Science and Technology, 42(1), 64–74, 2008.

Wagner, R., Kaufmann, J., Möhler, O., Saathoff, H., Schnaiter, M., Ullrich, R., & Leisner, T.: Heterogeneous ice nucleation ability of NaCl and sea salt aerosol particles at cirrus temperatures, Journal of Geophysical Research: Atmospheres, 123, 2841–2860. Doi: 10.1002/2017JD027864, 2018.

Bartels-Rausch, T., V. Bergeron, J. H. E. Cartwright, R. Escribano, J. L. Finney, H. Grothe, P. J. Gutíerrez, J. Haapala, W. F. Kuhs, J. B. C. Pettersson, S. D. Price, C. I. Sainz-Dıaz, D. J. Stokes, G. Strazzulla, E. S. Thomson, H. Trinks, and N. Uras-Aytemiz. Ice structures, patterns, and processes: A view across the icefields. Rev. Mod. Phys., 84:885–944, May 2012.

Bartels-Rausch, T., H.-W. Jacobi, T. F. Kahan, J. L. Thomas, E. S. Thomson, J. P. D. Abbatt, M. Ammann, J. R. Blackford, H. Bluhm, C. Boxe, F. Domine, M. M. Frey, I. Gladich, M. I. Guzmán, D. Heger, T. Huthwelker, P. Kľan, W. F. Kuhs, M. H. Kuo, S. Maus, S. G. Moussa, V. F. McNeill, J. T. Newberg, J. B. C. Pettersson, M. Roeselová, and J. R. Sodeau. A review of air–ice chemical and physical interactions (AICI): liquids, quasi-liquids, and solids in snow. Atmospheric Chemistry and Physics, 14(3):1587–1633, 2014.

Kirkevåg, A., T. Iversen, Ø. Seland, C. Hoose, J. E. Kristjánsson, H. Struthers, A. Ekman, S. Ghan, J. Griesfeller, D. Nilsson, and M. Schulz: Aerosol-climate interactions in the Norwegian Earth System Model - NorESM1-M, Geosci. Model Dev., 6, 207-244, doi:10.5194/gmd-6-207-2013, 2013.

Nilsson, E. D. and Rannik, Ü.: Turbulent aerosol fluxes over the Arctic Ocean: 1. Dry deposition over sea and pick ice, J. Geophys. Res., 106 (D23), 32125-32137, 2001.

Spracklen, D. V., Arnold, S. R., Sciare, J., Carslaw, K. S., and Pio, C.: Globally significant oceanic source of organic carbon aerosol, Geophys. Res. Lett., 35, L12811, doi:10.1029/2008GL033359, 2008.